# Sparse Training from Random Initialization:
# Aligning Lottery Ticket Masks using Weight Symmetry

Mohammed Adnan [* 1 2]   Rohan Jain [* 1]   Ekansh Sharma [3 2]   Rahul G. Krishnan [3 2]   Yani Ioannou [1]

## Abstract

The Lottery Ticket Hypothesis (LTH) suggests there exists a sparse LTH mask and weights that achieve the same generalization performance as the dense model while using significantly fewer parameters. However, finding a LTH solution is computationally expensive, and a LTH's sparsity mask does not generalize to other random weight initializations. Recent work has suggested that neural networks trained from random initialization find solutions within the same basin modulo permutation, and proposes a method to align trained models within the same loss basin. We hypothesize that misalignment of basins is the reason why LTH masks do not generalize to new random initializations and propose permuting the LTH mask to align with the new optimization basin when performing sparse training from a different random init. We empirically show a significant increase in generalization when sparse training from random initialization with the permuted mask as compared to using the non-permuted LTH mask, on multiple datasets (CIFAR-10/100 & ImageNet) and models (VGG11 & ResNet20/50). Our codebase for reproducing the results is publicly available at here.

## 1. Introduction

In recent years, foundation models have achieved state-of-the-art results for different tasks. However, the exponential increase in the size of state-of-the-art models requires a similarly exponential increase in the memory and computational costs required to train, store and use these models — decreasing the accessibility of these models for researchers

and practitioners alike. To overcome this issue, different model compression methods, such as pruning, quantization and knowledge distillation, have been proposed to reduce the model size at different phases of training or inference. Post-training model pruning (Han et al., 2016) has been shown to be effective in compressing the model size, and seminal works have demonstrated that large models can be pruned after training with minimal loss in accuracy (Gale et al., 2019; Han et al., 2015). While model pruning makes inference more efficient, it does not reduce the computational cost of training the model.

Motivated by the goal of training a sparse model from a random initialization, Frankle & Carbin (2019) demonstrated that training with a highly sparse mask is possible and proposed the Lottery Ticket Hypothesis (LTH) to identify sparse subnetworks that, when trained, can match the performance of a dense model. The key caveat is that a dense model must first be trained to find the sparse mask, which can *only* be used with the same random initialization that was used to train the dense model. Despite LTH seeing significant interest in the research community, LTH masks cannot be used to train from a new random initialization. Furthermore, it has been observed empirically that the LTH is impractical for finding a diverse set of solutions (Evci et al., 2022).

This posits our main research questions: *How can we train a LTH mask from a different random initialization while maintaining good generalization? Would doing so find a more diverse set of solutions than observed with the LTH itself?*

In this work, we try to understand why the LTH does not work for different random initializations from a weight-space symmetry perspective. Our hypothesis is that to reuse the LTH winning ticket mask with a different random initialization, the winning ticket mask obtained needs to be permuted such that it aligns with the optimization basin associated with the new random initialization. We illustrate our hypothesis in Figure 1.

To empirically validate our hypothesis, we obtain a sparse mask using Iterative Magnitude Pruning (IMP) (Renda et al., 2020; Han et al., 2015) on model $A$ (from Figure 1) and show that given a permutation that aligns the optimization basin of model $A$ and a new random initialization, the mask can

---
[*]Equal contribution [1] Schulich School of Engineering, University of Calgary [2]Vector Institute for AI [3]Dept. of Computer Science, University of Toronto. Correspondence to: Mohammed Adnan <adnan.ahmad@ucalgary.ca>, Yani Ioannou <yani.ioannou@ucalgary.ca>.

*Proceedings of the $42^{nd}$ International Conference on Machine Learning*, Vancouver, Canada. PMLR 267, 2025. Copyright 2025 by the author(s).

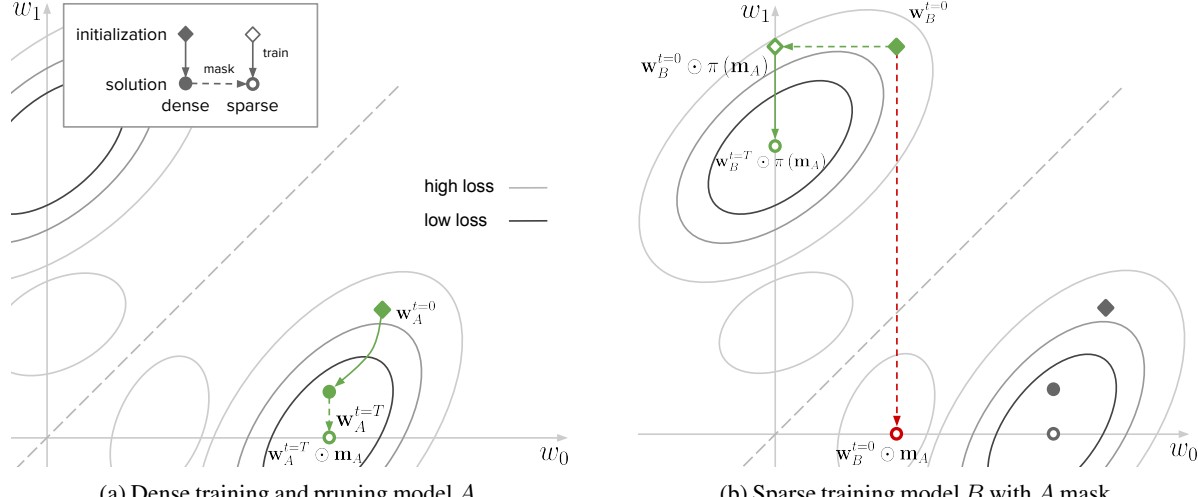

(a) Dense training and pruning model $A$.

(b) Sparse training model $B$ with $A$ mask.

*Figure 1.* **Weight Symmetry and the Sparse Training Problem**. A model with a single layer and only two parameters, $\mathbf{w} = (w_0, w_1)$, operating on a single input $x_0$ has weight symmetry in the 2D loss landscape as illustrated above. In (a) the original dense model, $\mathbf{w}_A$, is trained from a random dense initialization, $\mathbf{w}_A^{t=0}$ to a dense solution, $\mathbf{w}_A^{t=T}$, which is then pruned using weight magnitude resulting in the mask $\mathbf{m}_A = (1, 0)$. In (b), naively using the same mask to train a model, B, from a different random initialization will likely result in the initialization being far from a good solution. Permuting the mask to match the (symmetric) basin in which the new initialization is in will enable sparse training. See Figure 12 in Appendix E for the full figure also including the Lottery Ticket Hypothesis (LTH).

be reused. The sparse model (with the permuted mask) can be trained to closer match the generalization performance of the LTH solution, and the permuted mask improves the generalization of the trained sparse model compared to the non-permuted mask. Furthermore, we observe drastically increased functional diversity when using our approach compared to LTH solutions. Our contributions are as follows:

1. We hypothesize that the LTH (Frankle & Carbin, 2019) fails to generalize well with a new random initialization due to a mismatch between the optimization basin of the winning ticket mask and the new random initialization's solution basin. We propose a method based on permutation matching between two dense models, that permutes the winning ticket's sparse mask to align with the optimization basin of the new random initialization. We empirically demonstrate on CIFAR-10/100 and ImageNet datasets using VGG11 and ResNet models of varying widths that permuting the LTH sparse mask to align with the new random initialization improves the performance of the trained model (permuted), compared to the model trained without permuting the sparse mask (naive).

2. We show that models trained from random initialization using the permuted LTH mask are much more functionally diverse in the solutions they learn than those found from training the LTH winning ticket mask and initialization alone (Evci et al., 2022), across several existing functional diversity metrics and improved ensemble performance.

3. Furthermore, our experiments provide novel insights about the LTH and the corresponding dense model: we show that for a fixed initialization, the dense solution and the corresponding LTH solution remain in the same loss basin once we take into account *variance collapse*. Notably, our conclusion differs from the conclusion drawn by Paul et al. (2023), where they did not consider the variance collapse issue when interpolating between the sparse and dense solutions.

## 2. Background & Related Work

**Linear Mode Connectivity.** A pair of trained neural networks are said to be linearly connected if the loss along the linear path between the models remains small. The phenomenon of linear (mode) connectivity was first observed in the context of Stochastic Gradient Descent (SGD) by Nagarajan & Kolter (2019), where they showed that two neural networks trained from the same initialization but with different data orders exhibit linear connectivity. The term Linear-Mode Connectivity (LMC) was introduced by Frankle et al. (2020), where they showed that independently trained neural networks can be linearly connected.

**Linear Mode Connectivity *modulo* Permutation.** Entezari et al. (2022) further observed that while a model and its randomly permuted counterpart are functionally equivalent, they are rarely linearly connected in the weight space. This misalignment suggests the presence of *loss barriers* — regions along a linear path between models where the loss is

significantly higher than at the endpoints. They conjectured that independently obtained SGD solutions exhibit no loss barrier when accounting for permutation symmetries, suggesting that all SGD-trained networks converge to a single basin modulo permutations. Building on this conjecture, several algorithms have been developed to address permutation invariance by aligning trained networks to the same optimization basin (Ainsworth et al., 2023; Jordan et al., 2023; Singh & Jaggi, 2020; Tatro et al., 2020). Ainsworth et al. (2023) demonstrated that two models trained from different random initializations find solutions within the same basin modulo permutation symmetry. They proposed a permutation matching algorithm to permute the units of one model to align it with a reference model, enabling LMC (Frankle et al., 2020). The use of activation matching for model alignment was originally introduced by Li et al. (2015), to ensure models learn similar representations when performing the same task. Jordan et al. (2023) investigated the poor performance of interpolated networks, attributing it to a phenomenon they termed "*variance collapse*". To address this, they proposed a method that rescales the hidden units, leading to significant improvements in the generalization performance of interpolated networks. A rigorous study from Sharma et al. (2024) introduced a notion of *simultaneous weak linear connectivity* where a permutation, $\pi$, aligning two networks also simultaneously aligns two larger fully trained networks throughout the entire SGD trajectory and the same $\pi$ also aligns successive iterations of independently sparsified networks found via weight rewinding. Sharma et al. (2024) also showed that for certain neural networks, sparse mask obtained via weight rewinding can be reused modulo permutations without hurting the test performance.

**Lottery Ticket Hypothesis.** The LTH proposes to solve the sparse training problem by re-using the same initialization as used to train the pruned models. For very small models, training from such an initialization maintains the generalization performance of the pruned model and demonstrates that training with a highly sparse mask is possible (Frankle & Carbin, 2019). However, subsequent work has shown that obtaining winning tickets for modestly-sized models requires using *weight rewinding* (Frankle et al., 2020) — requiring significantly more compute than dense training alone, especially considering that LTH also requires IMP, i.e. training of iteratively sparsified models. We include a detailed description of IMP in Appendix A.3. Paul et al. (2023) analyzed the IMP algorithm and showed that sparse network obtained after $K^{\text{th}}$ IMP iteration is linearly connected to the sparse model obtained after $K+1^{\text{th}}$ IMP iteration. In this work, we show that once we take into account the *variance collapse* studied in Jordan et al. (2023), we are able to show that the sparse solution obtained after the $K^{\text{th}}$ iteration is linearly connected to the dense solution. Furthermore, recent work has shown that the LTH effectively

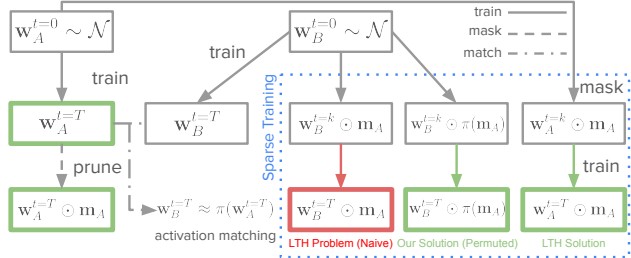

*Figure 2.* The overall framework of the training procedure, beginning with two distinct dense random weight initializations, $\mathbf{w}_A^{t=0}$, $\mathbf{w}_B^{t=0}$ sampled from a normal distribution, $\mathcal{N}$. The sparse training problem attempts to train the random initialization, $\mathbf{w}_B^{t=0}$ using the naive mask $\mathbf{m}_A$, found by pruning a dense trained model, $\mathbf{w}_A^{t=T}$. However, this results in poor generalization performance (Frankle et al., 2020). We propose to instead train $\mathbf{w}_B^{t=k}$ at some rewound epoch $k$, equipped with a *permuted* mask $\pi(\mathbf{m}_A)$. We show that this achieves more comparable generalization to the pruned model/trained LTH solution, $\mathbf{w}_A^{t=T} \odot \mathbf{m}_A$.

re-learns the original pruned solution it is derived from (Evci et al., 2022). To make any practical use of sparse training, finding methods of sparse training from random initialization is necessary to realize any efficiency gains in training.

**Weight Symmetry.** Hecht-Nielsen (1990) demonstrated that neural networks are *permutation invariant*, where swapping any two neurons within a hidden layer does not alter the underlying function being learned. The permuted network remains functionally equivalent to its original configuration, i.e. neural networks are symmetric functions. The existence of permutation symmetries in weight space creates copies of global minima at different points in weight space (Entezari et al., 2022; Goodfellow et al., 2016; Simsek et al., 2021). Weight sparsity, achieved through pruning, can reduce the number of weight symmetries in a neural network. Pruning neurons reduces the number of permutation symmetries in a layer. Unstructured pruning, in heterogeneously removing individual weights, can break the weight symmetry of individual neurons in layers.

## 3. Method

**Motivation.** In this work, we try to understand *why LTH masks fail to transfer to a new random initialization*. Our hypothesis is that the loss basin corresponding to the LTH mask is not aligned with the new random initialization, as shown in Figure 1. Since the sparse mask is not aligned with the basin of the new random initialization, sparse training does not work well; therefore, aligning the LTH mask with the new random initialization may improve sparse training and enable the transfer of LTH masks to new random initializations.

**Permutation Matching.** Ainsworth et al. (2023) showed that the permutation symmetries in the weight space can be leveraged to align the basin of two models trained from different random initializations. The permutation mapping can be obtained by either matching activations or weights. In this work, we use activation matching to obtain the permutation mapping as it has been shown to be more stable in recent works (Sharma et al., 2024). Activation matching tries to find a permutation mapping, $\pi \in S_d$ (where $S_d$ is the permutation group of order $d!$) such that by permuting the parameters of the second model, the correlation between the activations of the two models is maximized. For a model consisting of $L$ layers, each layer is sequentially matched and permuted starting from the input layer. Let $Z_l^A, Z_l^B \in \mathbb{R}^{d \times n}$ be the activations of layer $l$ of model $A$ and $B$ respectively obtained using the training data, where $d$ represents the dimensionality of the activations at layer $l$ and $n$ is the number of training data points. Then a permutation mapping for layer $l$, $\pi_l$, is obtained by solving:

$$\pi_l = \underset{\pi}{\mathrm{argmin}} ||Z_l^B - \pi Z_l^A||$$
$$= \underset{\pi}{\mathrm{argmax}} \langle \pi, Z^B (Z^A)^\top \rangle_F \quad (1)$$

where $\langle .,. \rangle_F$ denotes the Frobenius inner product. Equation (1) can be formulated as a Linear Assignment Problem (LAP) (Bertsekas, 1998; Ito et al., 2024) solved via the Hungarian algorithm (Kuhn, 2010); however, the permutation found is not global optima but a greedy/approximate solution as permutation matching is a NP-hard problem. Once the permutation mapping is obtained for all the layers, the model $A$ can be permuted to match model $B$. To ensure that the permuted model does not change functionally when permuting the output dimension of layer $l$, the input dimension of the next layer is also permuted accordingly. Let $W_l$ and $b_l$ be the weights and bias of layer $l$ respectively, then the permuted weight matrix $W_l^p$ and permuted bias $b_l^p$ for each layer can be mathematically represented as,

$$W_l^p = \pi_l W_l (\pi_{l-1})^\top, \qquad b_l^p = \pi_l b_l. \quad (2)$$

**Evaluating Permutation Matching.** Since LAP uses a greedy search to find an approximate solution, to ensure that the permuted model $A$ and model $B$ lie in the same basin, we evaluate the LMC (loss barrier) between the two models. More formally, let $\theta_1, \theta_2$ be the parameters of two networks, then the loss barrier $\mathcal{B}$ is defined as:

$$\mathcal{B}(\theta_1, \theta_2) := \sup_{\alpha \in [0,1]} \Big[ \mathcal{L}\big((1-\alpha)\theta_1 + \alpha\theta_2\big)$$
$$- \big((1-\alpha)\mathcal{L}(\theta_1) + \alpha\mathcal{L}(\theta_2)\big) \Big] \geq 0, \quad (3)$$

where $\mathcal{L}$ is the loss function evaluated on the training dataset. If $\mathcal{B}(\theta_1, \theta_2) \approx 0$, it is said that $\theta_1$ and $\theta_2$ are linearly mode connected.

To ensure that the permutation mapping, $\pi$, can closely match model $A$ and model $B$, we evaluate the loss barrier between the permuted model $A$ and model $B$. However, aligning neurons alone is not sufficient to establish a low loss barrier due to variance collapse (Jordan et al., 2023). To overcome the variance collapse issue, we used REPAIR (Jordan et al., 2023) to correct the variance of the activations in the interpolated/merged model. As shown in Figure 3, the loss barrier after permutation matching and correcting the variance (REPAIR) is lower than the loss at random initialization, showing permutation mapping can match the models to bring them closer/in the loss basin.

**Aligning Masks via Weight Symmetry.** In contrast to previous works (Ainsworth et al., 2023), we are interested in permuting the mask obtained by LTH such that the optimization basin of the permuted sparse mask and the new random initialization is aligned. To validate our hypothesis, we train two dense models, $\mathbf{w}_A^{t=0}$ and $\mathbf{w}_B^{t=0}$, where $t$ denotes the epoch, to convergence (trained for $T$ epochs) and then use activation matching (Jordan et al., 2023) to find the permutation mapping $\pi$, such that the activations of $\pi(\mathbf{w}_A^{t=T})$ and $\mathbf{w}_B^{t=T}$ are aligned. Mask $\mathbf{m}_A$, obtained using IMP, is also permuted with the same permutation map $\pi$. The intuition is that the permuted mask aligns with the loss basin of model $\mathbf{w}_B^{t=T}$, which is necessary for sparse training and, therefore, the sparse model can be more easily optimized (see Figure 2). We denote training with the permuted mask, $\pi(\mathbf{m}_A)$ as *permuted* and with the non-permuted mask, $\mathbf{m}_A$ as *naive*.

**Sparse Training.** We first show that the dense solution $\mathbf{w}_A^{t=T}$ and the LTH solution obtained by training a model with sparse mask $\mathbf{m}_A$ remain in the same linearly connected mode if one fixes the variance collapse identified by Jordan et al. (2023) by updating the activation statistics via the REPAIR method. We show in Figure 4a the error barrier after applying the REPAIR method remains considerably low as we increase sparsity by iteratively pruning (IMP). These results extend the findings of Paul et al. (2023) to show that when variance collapse is taken into account, the LTH solution remains in the same linearly connected basin as the original dense solution.

In sparse training, the model is trained with a mask $\mathbf{m}$, masking some of the weights, during both forward and backward passes. To evaluate the transferability of the permuted LTH mask we train, a different random initialization $\mathbf{w}_B^{t=0}$, the LTH sparse mask $\mathbf{m}_A$ and permuted LTH mask $\pi(\mathbf{m}_A)$, which we denote the naive and permuted solution respectively. We also evaluate the LTH baseline, i.e., training model $\mathbf{w}_A^{t=0}$ with mask $\mathbf{m}_A$. Since LTH requires weight rewinding to an earlier point in training, we also use a rewind checkpoint from epoch $t = k \ll T$ for both the baselines and permuted solution. Figure 4b shows that LTH, dense model $A$ and sparse permuted solutions all lie in the same mode.

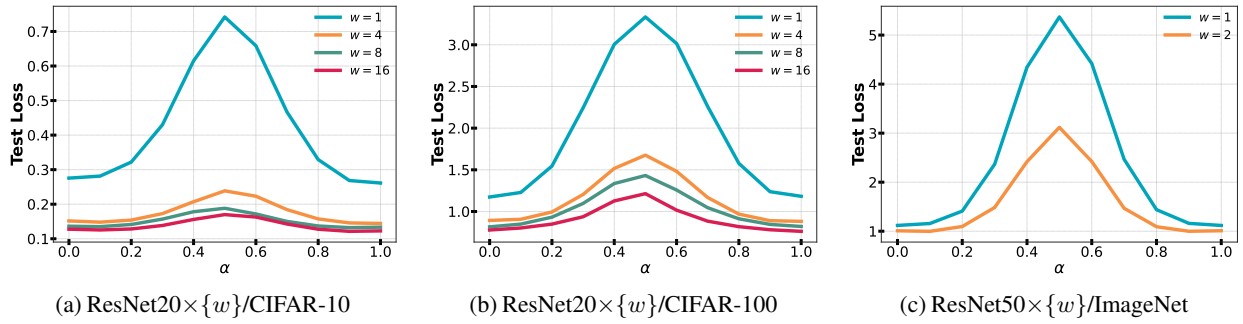

(a) ResNet20×{$w$}/CIFAR-10      (b) ResNet20×{$w$}/CIFAR-100      (c) ResNet50×{$w$}/ImageNet

*Figure 3.* **Larger width exhibits better LMC.** Plots showing linear interpolation between $\pi(\mathbf{w}_A^{t=T})$ and $\mathbf{w}_B^{t=T}$ where $\pi$ was obtained through activation matching between two dense models for varying widths, $w$. As the width of the model increases, the permutation matching algorithm gets more accurate, thereby reducing the loss barrier (i.e., better LMC), which is evaluated on the test set. This shows that the permutation matching can find a better mapping, $\pi$, for wider models, explaining why the permuted mask works better in the case of wider models.

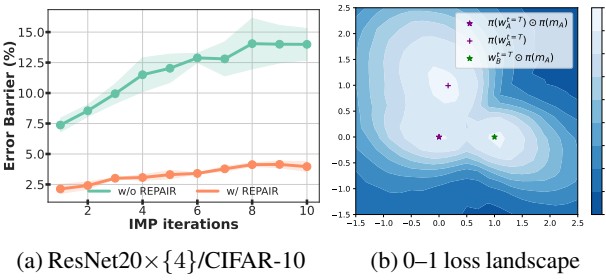

(a) ResNet20×{4}/CIFAR-10      (b) 0–1 loss landscape

*Figure 4.* **LTH solution remains in the same linearly connected mode as the dense solution.** In Figure 4a we plot the error barrier between the dense solution and the sparse solution (y-axis) vs the IMP iteration corresponding to the sparse solution (x-axis), for 90% sparsity. We observe that after fixing variance collapse via the REPAIR method, the error barrier between the dense and the sparse solutions remains small, thus showing that LTH solution remains in the same linearly connected mode as the dense solution. In Figure 4b we visualize the 0–1 loss landscape of ResNet20×{4}/CIFAR-10. The figure is generated by evaluating the 0–1 loss spanned by three models in the figure. We show that, modulo permutations, reusing the permuted mask leads to convergence in the same mode as the original model, i.e. the LTH solution. Hence, there is a small loss barrier between the permuted and LTH solutions, demonstrating they are within the same linearly connected mode.

# 4. Results

To validate our hypothesis, we trained ResNet20 (He et al., 2016) and VGG11 (Simonyan & Zisserman, 2015) models on the CIFAR-10/100 datasets (Krizhevsky, 2009) (details in Appendix A.1) across different levels of sparsity ($S = 0.80, 0.90, 0.95, 0.97$). We used ResNet20 with varying widths ($w = 1, 4, 8, 16$) to study the effect of increasing width on the permutation matching and, thereby, the performance of the permuted sparse model. We also demonstrate our hypothesis on the large-scale ImageNet dataset (Deng et al., 2009) using ResNet50, showing the efficacy of our method across different models and datasets of varying sizes.

## 4.1. Experimental Results.

**ResNet20/CIFAR-10 & CIFAR-100.** We trained ResNet20 on the CIFAR-10/100 datasets. As shown in Figures 5 and 6, the permuted solution outperforms the naive baseline across all model widths and rewind points. Since it is more difficult to train models with higher sparsity, the gap between naive and permuted solutions increases as sparsity increases, as shown in Figure 5d for width multiplier 1,4,8, and 16. It can also be observed that at higher sparsity increasing the rewind point improves both the LTH and permuted solution but not the naive solution. The improved performance of the permuted solution over naive supports our hypothesis and shows that misalignment of the LTH mask and loss basin corresponding to the new random initialization could explain why LTH masks do not transfer to different initializations. We also show accuracy vs. sparsity plots for $k = \{10, 25, 50, 100\}$ (details in Appendix A.5); as sparsity increases, the gap between permuted and naive solution increases for all rewind points. As illustrated in figure Figure 5, neither the LTH nor the permuted solution performs effectively with random initialization ($k = 0$) but improves on increasing the rewind point up to a certain point, beyond which it plateaus. Detailed results are presented in Tables 5 to 8 in Appendix A.4.

We also validated our hypothesis on CIFAR-100 using ResNet20 with varying widths. As shown in Figure 6, the permuted solution consistently outperforms the naive solution, showing that our hypothesis holds true across different models and datasets. Similar to the CIFAR-10 dataset, as we increase the model width multiplier, the gap between the permuted and naive solution increases, showing the efficacy of our method. Detailed results are presented in Tables 11 to 14 in Appendix A.4.

**VGG11/CIFAR-10.** We utilize the modified VGG11 architecture implemented by Jordan et al. (2023) trained on CIFAR-10 (details in Appendix A.1). We observe that for a moderate sparsity (80%) in Figure 8a, the gap

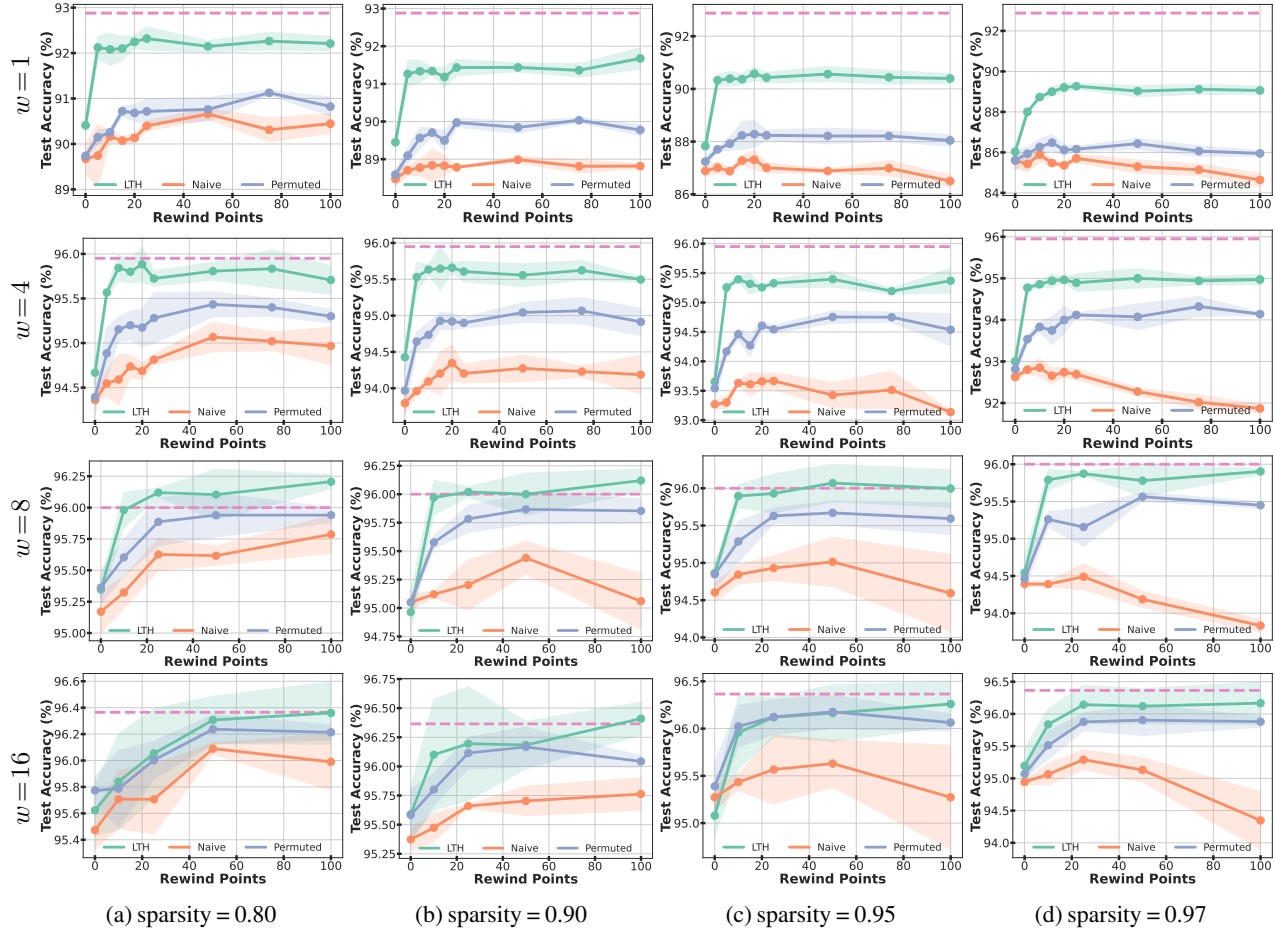

*Figure 5.* **ResNet20×{$w$}/CIFAR-10.**Test accuracy of sparse network solutions vs. increasing rewind points for different sparsity levels and widths, $w$. The dashed (**- -**) line shows the dense model accuracy. The effect of the rewind point on the test accuracy for different sparsities is shown. As the width increases, the gap between training from a random initialization with the permuted mask and the LTH/dense baseline (dashed line) decreases, unlike training with the non-permuted mask (naive), showing a model trained with the permuted mask generalizes better than naive.

between the permuted and the naive baseline is not large, however for a higher sparsity level (90%), the permuted solution significantly outperforms the naive solution as shown in Figure 8b. For the VGG11 model, on increasing the rewind point, the permuted solution closely matches the accuracy of LTH, while the naive solution significantly plateaus and does not improve on increasing the rewind point. For higher sparsities, the naive baseline was unstable in training as the modified VGG11 architecture does not have BatchNorm layers (Ioffe & Szegedy, 2015); we omit those results in the discussion for a fair comparison. Detailed results are presented in Table 9 in Appendix A.4.

**ResNet50/ImageNet.** We also validated our hypothesis on the ILSVRC 2012 (ImageNet) dataset, which consists of 1.28 million images across 1,000 classes (Deng et al., 2009). We used the ResNet50 model to evaluate the performance of the permuted mask at different sparsity levels. As

observed in Figure 7, the permuted solution outperforms the naive solution across all sparsity levels, showing that our hypothesis holds true on large-scale datasets as well. While the permuted solution performs better than the naive solution, there is still a significant gap between LTH and the permuted solution in the case of the ImageNet dataset as compared to the CIFAR-10/100 dataset. This could be due to permutation matching not being accurate enough, as only a small subset of the training dataset was used for activation matching. This can also be visualized in terms of the loss barrier in Figure 3c between the permuted model $A$ and model $B$; the loss barrier after permutation is more prominent compared to the CIFAR dataset (Figures 3a and 3b). Thus, the permutation mapping $\pi$ cannot match the models perfectly in the case of ImageNet since the permutation matching algorithm uses a greedy search algorithm to find the permutation mapping. However, given a better mapping, it may be possible to further improve the performance of the permuted solution as discussed

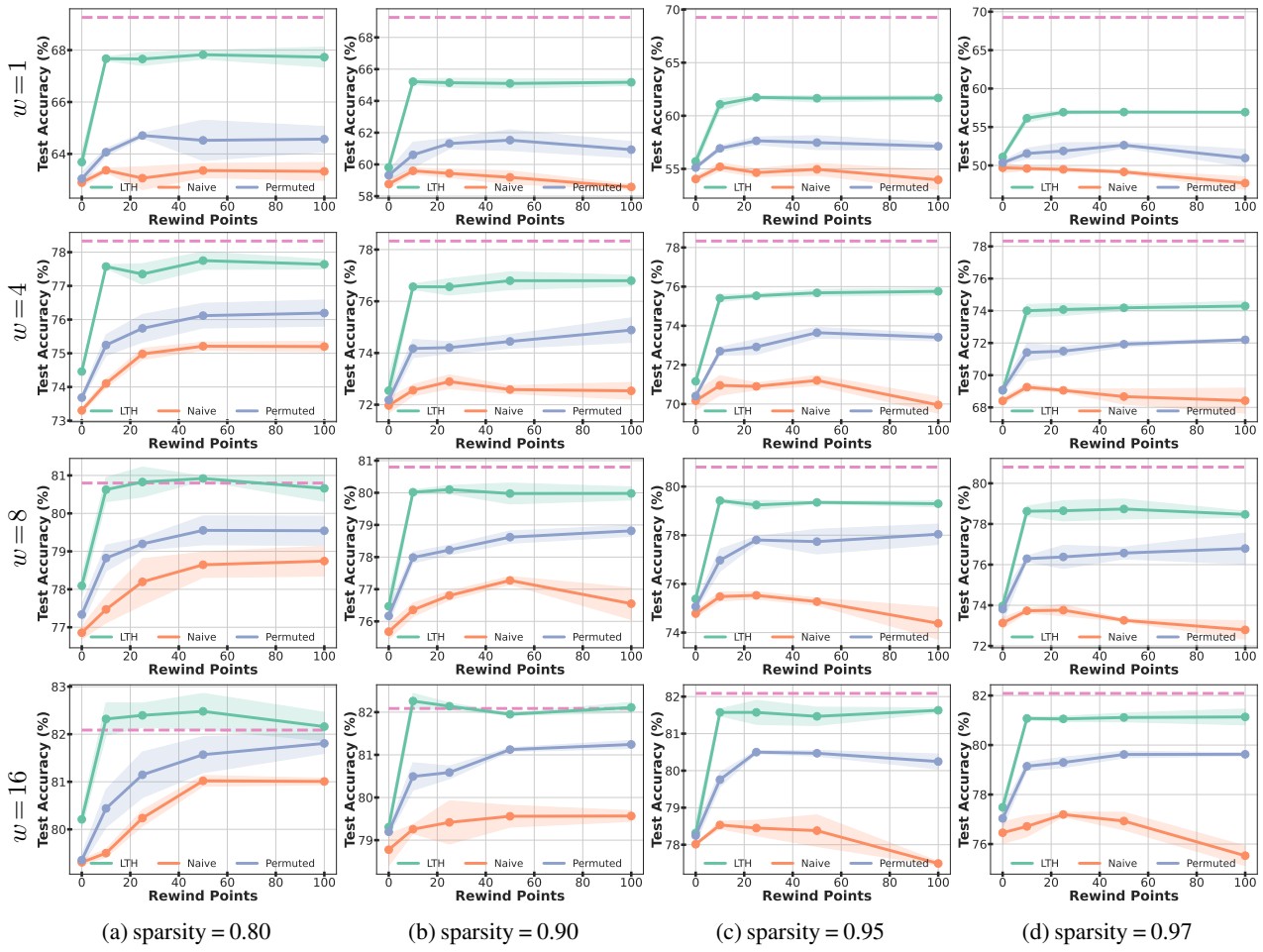

(a) sparsity = 0.80     (b) sparsity = 0.90     (c) sparsity = 0.95     (d) sparsity = 0.97

*Figure 6.* **ResNet20×{$w$}/CIFAR-100**. Test accuracy of sparse network solutions vs. increasing rewind points for different sparsity levels and widths, $w$. The dashed (**- -**) line shows the dense model accuracy. The effect of the rewind points on the test accuracy for different sparsities is shown. As the width increases, the gap between training from a random initialization with the permuted mask and the LTH/dense baseline (dashed line) decreases, unlike training with the non-permuted mask (naive), showing model trained with the permuted model generalizes better than naive.

in Section 4.3. Detailed results are presented in Table 10 in Appendix A.4. As demonstrated in Table 10, the permuted solution outperforms the naive approach by nearly 2% at higher sparsity levels.

### 4.2. Diversity Analysis of Permuted Models.

A limitation of LTH is that it consistently converges to very similar solutions to the original pruned model (Evci et al., 2022). Evci et al. (2022) speculate this occurs because the LTH is always trained with the same initialization/rewind point, and effectively relearns the same solution. Our hypothesis is that permuted LTH masks, trained with distinct initialization/rewind points and subject to approximation errors in permutation matching, may learn more diverse functions than the LTH itself. We analyze the diversity of sparse models trained at 90% sparsity, with either a permuted

mask (permuted), the LTH mask (naive), LTH mask & init. and the original pruned solution (IMP) on which the LTH is based. We follow the same analysis as Evci et al. (2022) and compare the diversity of the resulting models, over five different training runs, using disagreement score, KL divergence and JS divergence. We also compare with an ensemble of five models trained independently with different random seeds. As shown in Table 1, an ensemble of permuted models shows higher diversity across all the metrics than the LTH, showing that the permuted models learn a more diverse set of solutions. We provide additional details in Appendix C.

### 4.3. Effect of Model Width Multiplier.

Permutation matching is an NP-hard problem; the activation matching algorithm proposed by Ainsworth et al. (2023) does not find the global optimum; rather, it uses a greedy search to

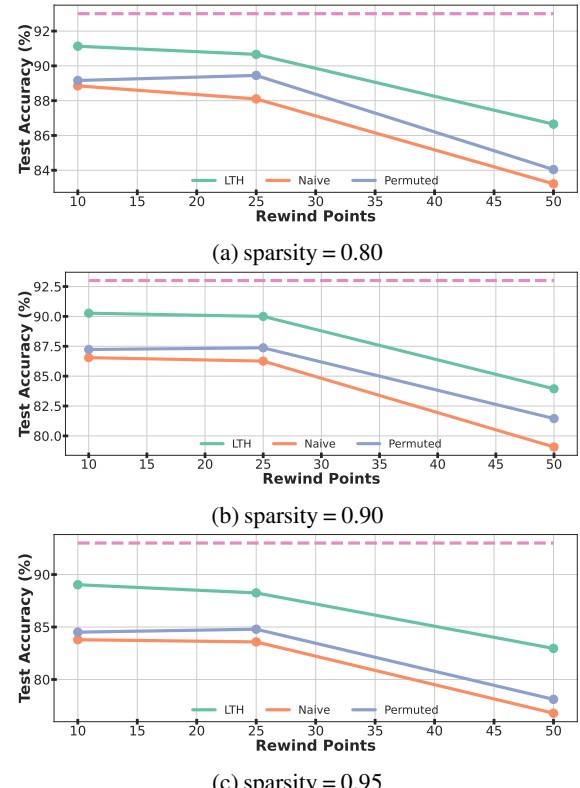

(a) sparsity = 0.80

(b) sparsity = 0.90

(c) sparsity = 0.95

*Figure 7.* **ResNet50×{1}/ImageNet**. Top-5 test accuracy vs. rewinds points of sparse network solutions at various sparsity levels. We observe the permuted solution consistently performing better than the naive solution for all sparsities. The dashed (- -) line shows the dense model accuracy.

explore a restricted solution space. Therefore, in practice, permutation matching does not perfectly align two models. However, it has been observed that for wider models, the algorithm can more closely align two models (Ainsworth et al., 2023; Sharma et al., 2024). To understand how the performance of the permuted model is affected by the approximation error of the matching algorithm, we evaluated the LMC and the accuracy of the permuted solution on ResNet20 models with varying layer widths. As shown in Figure 3, on increasing the layer width, the loss barrier of the interpolated network reduces, showing that permutation mapping becomes more accurate and aligns two models better. Also, it can be observed in Figures 5 and 6 that the permuted solution becomes close to the LTH solution on increasing the model width, showing that as the permutation matching becomes more accurate, the gap between the LTH and the permuted solution reduces.

## 5. Conclusion

In this work, we demonstrate new insights into sparse training from random initialization and the Lottery Ticket Hypothesis (LTH) by leveraging weight symmetry in Deep

*Table 1.* **Ensemble Diversity Metrics for CIFAR-10/CIFAR-100**. Although the mean test accuracy of LTH is higher, the ensemble of permuted models achieves better test accuracy due to better functional diversity of permuted models. Here we compare several measurements of function space similarity between the models including disagreement, which measures prediction differences (Fort et al., 2020), and Kullback–Leibler (KL)/Jenson-Shannon (JS) divergence, which quantify how much the output distributions of different models differ (Evci et al., 2022). As shown, the permuted masks achieve similar diversity as computational expensive IMP solutions, also resulting in ensembles with a similar increase in generalization.

| Mask | Test Accuracy (%) | Ensemble Acc. (%) | Disagreement | KL | JS |
|---|---|---|---|---|---|
| ResNet20×{1}/CIFAR-10 | | | | | |
| none (dense) | 92.76 ± 0.106 | - | - | - | - |
| IMP | 91.09 ± 0.041 | 93.25 | 0.093 | 0.352 | 0.130 |
| LTH | **91.15 ± 0.163** | 91.43 | 0.035 | 0.038 | 0.011 |
| permuted | 89.38 ± 0.170 | **91.75** | 0.107 | **0.273** | **0.091** |
| naive | 88.68 ± 0.205 | 91.07 | **0.113** | 0.271 | 0.089 |
| ResNet20×{4}/CIFAR-100 | | | | | |
| none (dense) | 78.37 ± 0.059 | - | - | - | - |
| IMP | 74.46 ± 0.321 | 79.27 | 0.259 | 1.005 | 0.372 |
| LTH | **75.35 ± 0.204** | 75.99 | 0.117 | 0.134 | 0.038 |
| permuted | 72.48 ± 0.356 | **77.85** | 0.278 | 0.918 | 0.327 |
| naive | 71.05 ± 0.366 | 76.15 | **0.290** | **0.970** | **0.348** |

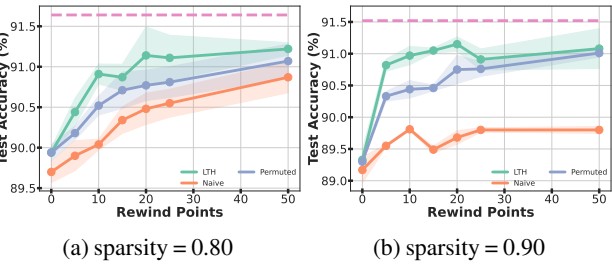

(a) sparsity = 0.80

(b) sparsity = 0.90

*Figure 8.* **VGG11×{1}/CIFAR-10.** Test accuracy of sparse solutions at increasing rewind points for different sparsity levels. The dashed (- -) line shows the dense model accuracy. In Figure 8b, the permuted solution closely matches the LTH solution. However, beyond a certain rewind point, i.e. for $k \geq 20$ the performance of the naive solution plateaus. Resulting in a more noticeable gap between the permuted and naive solutions compared to Figure 8a.

Neural Networks (DNNs). Our empirical findings across various models and datasets support the hypothesis that misalignment between the mask and loss basin prevents effective use of LTH masks with new initialization. Although finding a permutation to align dense models is computationally expensive, the goal of our work is to develop insights into the working of LTH and how the sparse mask can be reused, not to improve the efficiency of LTH. We hope that our work will spur future work in this direction and will be useful to the research community working in the realm of sparse training.

## Impact Statement

Our work focuses on improving sparse training and reducing the computational cost of training DNN. By reducing the computational and memory requirements of training and inference, sparse training can facilitate the deployment of deep learning models on resource-constrained devices. However, most current hardware cannot leverage unstructured sparsity, which currently limits the impact of this work. Furthermore, model pruning often introduces algorithmic bias in the model (Hooker et al., 2020), it is important to evaluate algorithmic bias of sparse models before deploying for real-world applications.

## Acknowledgements

We would like to acknowledge the assistance of Nayan Saxena with an early code prototype for the project, and Yigit Yargic with an early iteration of Figure 1.

We gratefully acknowledge the support of Alberta Innovates (ALLRP-577350-22, ALLRP-222301502), the Natural Sciences and Engineering Research Council of Canada (NSERC) (RGPIN-2022-03120, DGECR-2022-00358), and Defence Research and Development Canada (DGDND-2022-03120).

This research was enabled in part by support provided by the Digital Research Alliance of Canada (alliancecan.ca). Resources used in preparing this research were provided, in part, by the Province of Ontario, the Government of Canada through CIFAR, and companies sponsoring the Vector Institute.

MA is supported by the NSERC Postgraduate Scholarship, RBC Borealis through the Borealis AI Global Fellowship Award and the Digital Research Alliance of Canada EDIA Champions program. ES is supported by the Vector Research Grant at the Vector Institute. RGK gratefully acknowledges support from a Canada CIFAR AI Chair. YI is supported by a Schulich Research Chair.

## Contribution Statement

All authors contributed to the writing of the paper. MA and RJ implemented the method, conducted most experiments, and contributed the majority of the writing. MA and YI played key roles in designing the methodology and experimental setup. ES contributed to the design of the experimental setup, provided key insights, hypothesized the connection to the conclusion drawn by Paul et al. (2023), as summarized in Figure 4 and our contributions, and ran experiments to validate it. RGK and YI, as senior authors, provided feedback on the writing and methodology. YI conceptualized the research idea, designed Figures 1, 2 and 12, and also contributed feedback on code and experiments throughout the project.

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

# A. Appendix

## A.1. Implementation Details for ResNet20 & VGG11 on CIFAR-10 and CIFAR-100

**Architectures**    For residual neural networks, we train the standard ResNet20 on CIFAR-10 and CIFAR-100 with varying width. We implemented a scalar, $w$, that adjusts the number of channels in each convolutional and fully connected layer:

- **First Convolution Layer**: The number of output channels is scaled from 16 to $w \times 16$.

- **Layer 1,2,3**: The number of output channels for the convolutional blocks in these layers are scaled from 16, 32, and 64 to $w \times 16$, $w \times 32$, and $w \times 64$, respectively.

- **Fully Connected Layer**: The input dimension to the final linear layer is scaled to $w \times 64$.

For convolutional neural networks, we train a modified version of the standard VGG11 implemented by Jordan et al. (2023) on CIFAR-10. Primary differences are:

- A single fully connected layer at the end which directly maps the flattened feature map output from the convolutional layers to the 10 classes for CIFAR-10 classification.

- The classifier is set up for CIFAR-10 with 10 output classes as originally VGG11 was designed for ImageNet with 1000 output classes (Deng et al., 2009).

Each of our results for a given rewound point, $k$, is averaged over 3 runs.

**Datasets**    For our set of experiments we used the CIFAR-10 and CIFAR-100 datasets (Krizhevsky, 2009). We apply the following standard data augmentation techniques to the training set:

- `RandomHorizontalFlip`: Randomly flips the image horizontally with a given probability (by default, $50\%$).

- `RandomCrop`: Randomly crops the image to a size of $32 \times 32$ pixels, with a padding of 4 pixels around the image.

**Optimizers**    We use the following hyperparameters for ResNet20 and VGG11 trained on CIFAR-10/100, as outlined in Table 2.

| Hyperparameter | Value |
|---|---|
| Optimizer | SGD |
| Momentum | 0.9 |
| Dense Learning Rate | 0.08 |
| Sparse Learning Rate | 0.02 |
| Weight Decay | $5 \times 10^{-4}$ |
| Batch Size | 128 |
| Epochs ($T$) | 200 |

*Table 2.* Hyperparameters for dense and sparse training of both ResNet20 and VGG11.

## A.2. Implementation Details for ResNet50 on ImageNet

**Architecture**    We utilize the standard ResNet50 implementation provided by torchvision and customize PyTorch's distributed data parallel codebase for training models on ImageNet (Paszke et al., 2019).

**Dataset**    For our set of experiments we used the ImageNet dataset (Deng et al., 2009). We apply the following standard data augmentation techniques to the training set:

- `RandomHorizontalFlip`: Randomly flips the image horizontally with a given probability (by default, $50\%$).

- `RandomResizedCrop`: Randomly crops a region from the image and resizes it to $224 \times 224$ pixels.

**Optimizers**    We use the following hyperparameters for ResNet50 trained on ImageNet, as outlined in Table 3.

| Hyperparameter | Value |
|---|---|
| Optimizer | SGD |
| Momentum | 0.9 |
| Dense Learning Rate | 0.4 |
| Sparse Learning Rate | 0.4 |
| Weight Decay | $1 \times 10^{-4}$ |
| Batch Size | 1024 |
| Epochs ($T$) | 80 |

*Table 3.* Hyperparameters for dense and sparse training of ResNet50.

## A.3. Pruning

We apply standard Iterative Magnitude Pruning - Fine Tuning (IMP-FT) (Frankle & Carbin, 2019; Han et al., 2015; Renda et al., 2020) to obtain our final mask, $\mathbf{m}_A$, producing a sparse subnetwork $\mathbf{w}_A^{t=T} \odot \mathbf{m}_A$. For pruning, we utilize PyTorch's `torch.nn.utils.prune` library (Paganini & Forde, 2020).

1. In an unstructured, global manner, we identify and mask (set to zero) the smallest 20% of unpruned weights based on their magnitude.

2. This process is repeated for $s$ rounds to achieve the target sparsity $S$, with each subsequent round pruning 20% of the remaining weights.

3. During each round, the model is trained for `train_epochs_per_prune` epochs.

| Hyperparameter | ResNet20/VGG11 | ResNet50 |
|---|---|---|
| `train_epochs_per_prune` | 50 | 20 |
| Learning Rate | 0.01 | 0.04 |

*Table 4.* Hyperparameters used for pruning ResNet20/VGG11 on CIFAR-10/100 and ResNet50 on ImageNet.

## A.4. Results

Detailed results for ResNet20$\times\{w\}$/CIFAR-10 are provided in Tables 5 to 8, for VGG11$\times\{1\}$/CIFAR-10 in Table 9, for ResNet50$\times\{1\}$/ImageNet in Table 10, and for ResNet20$\times\{w\}$/CIFAR-100 in Tables 11 to 14.

## A.5. Additional Plots

Refer to Figures 9 and 10 for additional accuracy-vs-sparsity plots for ResNet20 on CIFAR-10 and CIFAR-100. Refer to Figure 11 for Top-1 accuracy vs. rewind points for ResNet50 on ImageNet.

# B. Early Matching

In our current methodology, we train both models $A$ and $B$ to convergence, resulting in the weight configurations $\mathbf{w}_A^{t=T}$ and $\mathbf{w}_B^{t=T}$, respectively. However, it has been observed in (Sharma et al., 2024) that it is possible to find permutation mapping earlier in training. In effort to reduce the computational cost associated with our approach, we aim to find a permutation $\pi$ that allows us to suitably align the weights of model $A$ at convergence, $\mathbf{w}_A^{t=T}$, with the weights of model $B$ at an earlier training iteration $i \ll T$, $\mathbf{w}_B^{t=i}$. In Table 15, we provide additional results on early matching with the CIFAR-10 dataset, which shows that models can be matched earlier in the training, thus reducing the computational cost of our method.

*Table 5.* **ResNet20×{1}/CIFAR-10**. Results using the ResNet20×{1} trained on CIFAR-10, from a rewind point $k$, using various methods of sparse training with sparsity $S$. LTH trains within the original dense/pruned solution basin, while naive/permuted train from a new random initialization.

| $S$ | Method | Rewind Epoch $k$ | | | | | | | | |
| --- | --- | --- | --- | --- | --- | --- | --- | --- | --- | --- |
| | | $k=0$ | 5 | 10 | 15 | 20 | 25 | 50 | 75 | 100 |
| 80% | LTH | $90.41 \pm 0.14$ | $92.12 \pm 0.25$ | $92.08 \pm 0.36$ | $92.10 \pm 0.27$ | $92.25 \pm 0.14$ | $92.32 \pm 0.26$ | $92.15 \pm 0.13$ | $92.26 \pm 0.19$ | $92.21 \pm 0.16$ |
| | naive | $89.67 \pm 0.35$ | $89.74 \pm 0.69$ | $90.16 \pm 0.14$ | $90.07 \pm 0.09$ | $90.13 \pm 0.11$ | $90.40 \pm 0.11$ | $90.66 \pm 0.12$ | $90.31 \pm 0.27$ | $90.45 \pm 0.22$ |
| | perm. | $89.74 \pm 0.05$ | $90.15 \pm 0.16$ | $90.26 \pm 0.08$ | $90.72 \pm 0.12$ | $90.68 \pm 0.18$ | $90.72 \pm 0.28$ | $90.76 \pm 0.27$ | $\mathbf{91.13 \pm 0.06}$ | $90.82 \pm 0.21$ |
| 90% | LTH | $89.45 \pm 0.10$ | $91.27 \pm 0.37$ | $91.34 \pm 0.29$ | $91.34 \pm 0.09$ | $91.18 \pm 0.27$ | $91.43 \pm 0.22$ | $91.44 \pm 0.12$ | $91.36 \pm 0.18$ | $91.68 \pm 0.28$ |
| | naive | $88.47 \pm 0.21$ | $88.70 \pm 0.14$ | $88.77 \pm 0.21$ | $88.84 \pm 0.43$ | $88.83 \pm 0.27$ | $88.78 \pm 0.02$ | $88.99 \pm 0.08$ | $88.81 \pm 0.17$ | $88.82 \pm 0.07$ |
| | perm. | $88.59 \pm 0.11$ | $89.09 \pm 0.22$ | $89.56 \pm 0.28$ | $89.71 \pm 0.12$ | $89.50 \pm 0.27$ | $89.97 \pm 0.13$ | $89.84 \pm 0.15$ | $\mathbf{90.03 \pm 0.07}$ | $89.77 \pm 0.15$ |
| 95% | LTH | $87.83 \pm 0.38$ | $90.33 \pm 0.22$ | $90.39 \pm 0.28$ | $90.37 \pm 0.21$ | $90.58 \pm 0.26$ | $90.43 \pm 0.20$ | $90.56 \pm 0.29$ | $90.44 \pm 0.26$ | $90.40 \pm 0.19$ |
| | naive | $86.89 \pm 0.21$ | $87.01 \pm 0.23$ | $86.88 \pm 0.13$ | $87.28 \pm 0.19$ | $87.31 \pm 0.36$ | $87.00 \pm 0.19$ | $86.88 \pm 0.08$ | $86.99 \pm 0.29$ | $86.50 \pm 0.22$ |
| | perm. | $87.24 \pm 0.22$ | $87.70 \pm 0.08$ | $87.92 \pm 0.25$ | $88.23 \pm 0.52$ | $\mathbf{88.29 \pm 0.52}$ | $88.24 \pm 0.20$ | $88.21 \pm 0.30$ | $88.21 \pm 0.20$ | $88.04 \pm 0.22$ |
| 97% | LTH | $86.03 \pm 0.22$ | $88.00 \pm 0.02$ | $88.73 \pm 0.05$ | $89.00 \pm 0.24$ | $89.21 \pm 0.23$ | $89.27 \pm 0.14$ | $89.03 \pm 0.27$ | $89.12 \pm 0.25$ | $89.06 \pm 0.21$ |
| | naive | $85.60 \pm 0.38$ | $85.43 \pm 0.40$ | $85.89 \pm 0.37$ | $85.48 \pm 0.13$ | $85.36 \pm 0.14$ | $85.70 \pm 0.21$ | $85.30 \pm 0.32$ | $85.14 \pm 0.29$ | $84.64 \pm 0.34$ |
| | perm. | $85.61 \pm 0.48$ | $85.93 \pm 0.34$ | $86.26 \pm 0.40$ | $\mathbf{86.48 \pm 0.39}$ | $86.12 \pm 0.27$ | $86.16 \pm 0.14$ | $86.43 \pm 0.27$ | $86.06 \pm 0.26$ | $85.95 \pm 0.14$ |

*Table 6.* **ResNet20×{4}/CIFAR-10**. Results using the ResNet20×{4} trained on CIFAR-10, from a rewind point $k$, using various methods of sparse training with sparsity $S$. LTH trains within the original dense/pruned solution basin, while naive/permuted train from a new random initialization. Note this table is the same setting as Table 5 except $w=4$.

| $S$ | Method | Rewind Epoch $k$ | | | | | | | | |
| --- | --- | --- | --- | --- | --- | --- | --- | --- | --- | --- |
| | | $k=0$ | 5 | 10 | 15 | 20 | 25 | 50 | 75 | 100 |
| 80% | LTH | $94.67 \pm 0.14$ | $95.57 \pm 0.05$ | $95.84 \pm 0.15$ | $95.80 \pm 0.12$ | $95.88 \pm 0.20$ | $95.72 \pm 0.09$ | $95.81 \pm 0.10$ | $95.83 \pm 0.21$ | $95.71 \pm 0.16$ |
| | naive | $94.36 \pm 0.04$ | $94.55 \pm 0.14$ | $94.59 \pm 0.29$ | $94.74 \pm 0.13$ | $94.69 \pm 0.09$ | $94.81 \pm 0.06$ | $95.07 \pm 0.17$ | $95.02 \pm 0.11$ | $94.97 \pm 0.21$ |
| | perm. | $94.39 \pm 0.19$ | $94.88 \pm 0.28$ | $95.15 \pm 0.14$ | $95.20 \pm 0.16$ | $95.17 \pm 0.21$ | $95.28 \pm 0.29$ | $\mathbf{95.43 \pm 0.14}$ | $95.40 \pm 0.10$ | $95.30 \pm 0.08$ |
| 90% | LTH | $94.43 \pm 0.17$ | $95.53 \pm 0.21$ | $95.63 \pm 0.07$ | $95.65 \pm 0.30$ | $95.66 \pm 0.07$ | $95.61 \pm 0.14$ | $95.56 \pm 0.16$ | $95.62 \pm 0.14$ | $95.50 \pm 0.04$ |
| | naive | $93.79 \pm 0.15$ | $93.96 \pm 0.05$ | $94.09 \pm 0.11$ | $94.20 \pm 0.29$ | $94.35 \pm 0.25$ | $94.20 \pm 0.13$ | $94.27 \pm 0.19$ | $94.23 \pm 0.08$ | $94.19 \pm 0.27$ |
| | perm. | $93.97 \pm 0.29$ | $94.64 \pm 0.13$ | $94.73 \pm 0.17$ | $94.93 \pm 0.12$ | $94.92 \pm 0.11$ | $94.90 \pm 0.07$ | $95.04 \pm 0.14$ | $\mathbf{95.07 \pm 0.18}$ | $94.91 \pm 0.19$ |
| 95% | LTH | $93.65 \pm 0.12$ | $95.26 \pm 0.08$ | $95.39 \pm 0.05$ | $95.32 \pm 0.18$ | $95.26 \pm 0.03$ | $95.33 \pm 0.07$ | $95.40 \pm 0.14$ | $95.19 \pm 0.05$ | $95.37 \pm 0.21$ |
| | naive | $93.27 \pm 0.07$ | $93.30 \pm 0.11$ | $93.63 \pm 0.04$ | $93.61 \pm 0.21$ | $93.66 \pm 0.13$ | $93.67 \pm 0.14$ | $93.43 \pm 0.21$ | $93.51 \pm 0.32$ | $93.14 \pm 0.03$ |
| | perm. | $93.54 \pm 0.24$ | $94.17 \pm 0.07$ | $94.46 \pm 0.10$ | $94.27 \pm 0.19$ | $94.61 \pm 0.07$ | $94.54 \pm 0.07$ | $\mathbf{94.75 \pm 0.11}$ | $\mathbf{94.75 \pm 0.09}$ | $94.54 \pm 0.27$ |
| 97% | LTH | $93.00 \pm 0.11$ | $94.77 \pm 0.09$ | $94.86 \pm 0.06$ | $94.94 \pm 0.17$ | $94.96 \pm 0.06$ | $94.89 \pm 0.21$ | $95.00 \pm 0.24$ | $94.94 \pm 0.10$ | $94.97 \pm 0.13$ |
| | naive | $92.63 \pm 0.12$ | $92.80 \pm 0.10$ | $92.85 \pm 0.21$ | $92.66 \pm 0.21$ | $92.74 \pm 0.11$ | $92.69 \pm 0.14$ | $92.28 \pm 0.09$ | $92.02 \pm 0.18$ | $91.87 \pm 0.10$ |
| | perm. | $92.81 \pm 0.27$ | $93.54 \pm 0.08$ | $93.83 \pm 0.12$ | $93.75 \pm 0.34$ | $94.00 \pm 0.33$ | $94.12 \pm 0.04$ | $94.07 \pm 0.31$ | $\mathbf{94.32 \pm 0.24}$ | $94.14 \pm 0.04$ |

# C. Additional Details for Section 4.2

In Section 4.2, the IMP solution is trained independently over 5 different seeds with iterative pruning to obtain 5 different sparse/pruned solutions with different sparse masks/topologies ($M_1, M_2, M_3, M_4, M_5$). The LTH ensemble is trained using the same mask ($M_1$) and initialization ($w_1$) over 5 different runs (with different data order). Random initialization $w_1$ defines the winning ticket for mask $M_1$. The permuted ensemble is trained using 5 different permutations ($\pi_1, \pi_2, \pi_3, \pi_4, \pi_5$) of the same mask ($M_1$) with five different random weight initializations ($w_1, w_2, w_3, w_4, w_5$).

# D. Computational Overhead of the Permuted Solution

The primary difference in computational complexity between the LTH, naive, and permuted solutions lies in the process of neuronal alignment, where weight/activation matching is used to locate permutations in order to bring the hidden units of two networks into alignment. To obtain the permuted solution, two distinct models must be trained independently to convergence, after which their weights or activations are aligned through a permutation-matching process. This alignment, though relatively efficient, adds a small computational overhead compared to LTH and naive solutions, which do not involve matching steps. However, it's important to note that the primary goal of this study is not to improve training efficiency but rather to investigate why the LTH framework fails when applied to sparse training from new random initializations (not

*Table 7.* **ResNet20×{8}/CIFAR-10**. Results using the ResNet20×{8} trained on CIFAR-10, from a rewind point $k$, using various methods of sparse training with sparsity $S$. LTH trains within the original dense/pruned solution basin, while naive/permuted train from a new random initialization. Note this table is the same setting as Table 5 except $w=8$.

| $S$ | Method | $k=0$ | 10 | 25 | 50 | 100 |
|---|---|---|---|---|---|---|
| | | | | Rewind Epoch $k$ | | |
| 80% | LTH | $95.35 \pm 0.07$ | $95.98 \pm 0.14$ | $96.12 \pm 0.04$ | $96.10 \pm 0.20$ | $96.21 \pm 0.06$ |
| | naive | $95.17 \pm 0.17$ | $95.32 \pm 0.13$ | $95.63 \pm 0.13$ | $95.62 \pm 0.08$ | $95.79 \pm 0.15$ |
| | perm. | $95.36 \pm 0.14$ | $95.60 \pm 0.15$ | $95.89 \pm 0.19$ | $\mathbf{95.94 \pm 0.17}$ | $\mathbf{95.94 \pm 0.06}$ |
| 90% | LTH | $94.96 \pm 0.18$ | $95.97 \pm 0.15$ | $96.02 \pm 0.05$ | $96.00 \pm 0.19$ | $96.12 \pm 0.10$ |
| | naive | $95.05 \pm 0.07$ | $95.12 \pm 0.03$ | $95.20 \pm 0.22$ | $95.44 \pm 0.14$ | $95.06 \pm 0.25$ |
| | perm. | $95.05 \pm 0.05$ | $95.58 \pm 0.06$ | $95.78 \pm 0.12$ | $\mathbf{95.87 \pm 0.13}$ | $95.85 \pm 0.11$ |
| 95% | LTH | $94.86 \pm 0.08$ | $95.90 \pm 0.15$ | $95.93 \pm 0.26$ | $96.07 \pm 0.25$ | $96.00 \pm 0.25$ |
| | naive | $94.60 \pm 0.14$ | $94.84 \pm 0.13$ | $94.93 \pm 0.17$ | $95.01 \pm 0.33$ | $94.59 \pm 0.52$ |
| | perm. | $94.85 \pm 0.19$ | $95.29 \pm 0.27$ | $95.63 \pm 0.11$ | $\mathbf{95.67 \pm 0.16}$ | $95.59 \pm 0.22$ |
| 97% | LTH | $94.54 \pm 0.23$ | $95.79 \pm 0.14$ | $95.87 \pm 0.03$ | $95.78 \pm 0.21$ | $95.90 \pm 0.04$ |
| | naive | $94.39 \pm 0.04$ | $94.39 \pm 0.04$ | $94.49 \pm 0.18$ | $94.19 \pm 0.11$ | $93.83 \pm 0.08$ |
| | perm. | $94.46 \pm 0.14$ | $95.26 \pm 0.10$ | $95.16 \pm 0.26$ | $\mathbf{95.56 \pm 0.06}$ | $95.45 \pm 0.05$ |

*Table 8.* **ResNet20×{16}/CIFAR-10**. Results using the ResNet20×{8} trained on CIFAR-10, from a rewind point $k$, using various methods of sparse training with sparsity $S$. LTH trains within the original dense/pruned solution basin, while naive/permuted train from a new random initialization. Note this table is the same setting as Table 5 except $w=16$.

| $S$ | Method | $k=0$ | 10 | 25 | 50 | 100 |
|---|---|---|---|---|---|---|
| | | | | Rewind Epoch $k$ | | |
| 80% | LTH | $95.62 \pm 0.19$ | $95.84 \pm 0.36$ | $96.05 \pm 0.34$ | $96.31 \pm 0.18$ | $96.36 \pm 0.24$ |
| | naive | $95.47 \pm 0.15$ | $95.71 \pm 0.22$ | $95.71 \pm 0.26$ | $96.09 \pm 0.04$ | $95.99 \pm 0.21$ |
| | perm. | $95.77 \pm 0.11$ | $95.79 \pm 0.29$ | $96.00 \pm 0.14$ | $\mathbf{96.24 \pm 0.11}$ | $96.21 \pm 0.06$ |
| 90% | LTH | $95.59 \pm 0.22$ | $96.10 \pm 0.48$ | $96.19 \pm 0.49$ | $96.18 \pm 0.20$ | $96.41 \pm 0.14$ |
| | naive | $95.37 \pm 0.09$ | $95.47 \pm 0.13$ | $95.66 \pm 0.01$ | $95.70 \pm 0.13$ | $95.76 \pm 0.14$ |
| | perm. | $95.58 \pm 0.22$ | $95.80 \pm 0.14$ | $96.11 \pm 0.13$ | $\mathbf{96.17 \pm 0.17}$ | $96.04 \pm 0.05$ |
| 95% | LTH | $95.08 \pm 0.21$ | $95.96 \pm 0.39$ | $96.12 \pm 0.21$ | $96.16 \pm 0.30$ | $96.26 \pm 0.23$ |
| | naive | $95.27 \pm 0.13$ | $95.43 \pm 0.09$ | $95.57 \pm 0.37$ | $95.63 \pm 0.25$ | $95.27 \pm 0.55$ |
| | perm. | $95.39 \pm 0.26$ | $96.02 \pm 0.22$ | $96.12 \pm 0.18$ | $\mathbf{96.18 \pm 0.18}$ | $96.06 \pm 0.09$ |
| 97% | LTH | $95.19 \pm 0.27$ | $95.84 \pm 0.25$ | $96.14 \pm 0.30$ | $96.12 \pm 0.27$ | $96.17 \pm 0.33$ |
| | naive | $94.94 \pm 0.04$ | $95.06 \pm 0.17$ | $95.29 \pm 0.15$ | $95.13 \pm 0.19$ | $94.35 \pm 0.45$ |
| | perm. | $95.07 \pm 0.06$ | $95.51 \pm 0.22$ | $95.88 \pm 0.14$ | $\mathbf{95.90 \pm 0.24}$ | $95.88 \pm 0.09$ |

associated with the winning ticket's mask).

# E. Full Symmetry Figure including Lottery Ticket Hypothesis

In Figure 12 we include the full version of Figure 1, including an illustration of the LTH in Figure 12b.

*Table 9.* **VGG11**×{1}**/CIFAR-10**. Results using the VGG11 trained on CIFAR-10, from a rewind point $k$, using various methods of sparse training with sparsity $S$. LTH trains within the original dense/pruned solution basin, while naive/permuted train from a new random initialization.

| | | Rewind Epoch $k$ | | | | | | |
|---|---|---|---|---|---|---|---|---|
| $S$ | Method | $k=0$ | 5 | 10 | 15 | 20 | 25 | 50 |
| 80% | LTH | $89.94 \pm 0.06$ | $90.44 \pm 0.17$ | $90.91 \pm 0.12$ | $90.87 \pm 0.16$ | $91.14 \pm 0.28$ | $91.11 \pm 0.08$ | $91.22 \pm 0.08$ |
| | naive | $89.70 \pm 0.13$ | $89.90 \pm 0.18$ | $90.04 \pm 0.07$ | $90.34 \pm 0.16$ | $90.48 \pm 0.19$ | $90.55 \pm 0.17$ | $90.87 \pm 0.19$ |
| | perm. | $89.94 \pm 0.1$ | $90.18 \pm 0.08$ | $90.52 \pm 0.17$ | $90.71 \pm 0.22$ | $90.77 \pm 0.19$ | $90.81 \pm 0.19$ | $\mathbf{91.07 \pm 0.21}$ |
| 90% | LTH | $89.33 \pm 0.16$ | $90.82 \pm 0.09$ | $90.97 \pm 0.14$ | $91.05 \pm 0.04$ | $91.15 \pm 0.11$ | $90.91 \pm 0.17$ | $91.08 \pm 0.31$ |
| | naive | $89.17 \pm 0.2$ | $89.55 \pm 0.02$ | $89.81 \pm 0.02$ | $89.49 \pm 0.05$ | $89.68 \pm 0.11$ | $89.80 \pm 0.03$ | $89.80 \pm 0.05$ |
| | perm. | $89.30 \pm 0.02$ | $90.33 \pm 0.08$ | $90.44 \pm 0.14$ | $90.46 \pm 0.04$ | $90.75 \pm 0.22$ | $90.76 \pm 0.12$ | $\mathbf{91.01 \pm 0.06}$ |

*Table 10.* **ResNet50**×{1}**/ImageNet**. Top-1 and Top-5 Accuracies of ResNet50×{1} trained on ImageNet, from a rewind point $k$, using various methods of sparse training with sparsity $S$.

| | | Top-1 Accuracy | | | Top-5 Accuracy | | |
|---|---|---|---|---|---|---|---|
| $S$ | Method | $k=10$ | 25 | 50 | $k=10$ | 25 | 50 |
| 80% | LTH | 72.87 | 72.16 | 65.23 | 91.13 | 90.66 | 86.65 |
| | naive | 69.13 | 68.94 | 60.30 | 88.85 | 88.1 | 83.22 |
| | perm. | **69.87** | 69.85 | 61.14 | 89.16 | **89.45** | 84.04 |
| 90% | LTH | 71.40 | 70.74 | 60.62 | 90.27 | 90.00 | 83.94 |
| | naive | 65.49 | 64.77 | 54.46 | 86.55 | 86.26 | 79.07 |
| | perm. | 66.25 | **66.37** | 57.40 | 87.23 | **87.37** | 81.45 |
| 95% | LTH | 68.61 | 68.07 | 59.83 | 89.03 | 88.25 | 82.96 |
| | naive | 61.39 | 60.77 | 51.78 | 83.79 | 83.58 | 76.79 |
| | perm. | 62.48 | **62.77** | 52.98 | 84.51 | **84.79** | 78.11 |

*Table 11.* **ResNet20$\times$\{1\}/CIFAR-100**. Results using the ResNet20$\times$\{1\} trained on CIFAR-100, from a rewind point $k$, using various methods of sparse training with sparsity $S$. LTH trains within the original dense/pruned solution basin, while naive/permuted train from a new random initialization.

| $S$ | Method | Rewind Epoch $k$ | | | | |
| --- | --- | --- | --- | --- | --- | --- |
| | | $k$=0 | 10 | 25 | 50 | 100 |
| 80% | LTH | $63.69 \pm 0.41$ | $67.67 \pm 0.08$ | $67.66 \pm 0.25$ | $67.82 \pm 0.17$ | $67.73 \pm 0.38$ |
| | naive | $62.89 \pm 0.16$ | $63.37 \pm 0.09$ | $63.07 \pm 0.44$ | $63.36 \pm 0.27$ | $63.33 \pm 0.35$ |
| | perm. | $63.04 \pm 0.24$ | $64.07 \pm 0.15$ | $\mathbf{64.71 \pm 0.10}$ | $64.52 \pm 0.78$ | $64.57 \pm 0.49$ |
| 90% | LTH | $59.81 \pm 0.29$ | $65.21 \pm 0.17$ | $65.15 \pm 0.28$ | $65.10 \pm 0.30$ | $65.17 \pm 0.21$ |
| | naive | $58.77 \pm 0.28$ | $59.59 \pm 0.18$ | $59.44 \pm 0.27$ | $59.19 \pm 0.41$ | $58.58 \pm 0.16$ |
| | perm. | $59.32 \pm 0.32$ | $60.60 \pm 0.79$ | $61.32 \pm 0.33$ | $\mathbf{61.53 \pm 0.65}$ | $60.93 \pm 0.51$ |
| 95% | LTH | $55.71 \pm 0.52$ | $61.08 \pm 0.54$ | $61.73 \pm 0.18$ | $61.65 \pm 0.37$ | $61.68 \pm 0.18$ |
| | naive | $54.04 \pm 0.29$ | $55.20 \pm 0.39$ | $54.65 \pm 0.38$ | $54.96 \pm 0.57$ | $53.97 \pm 0.91$ |
| | perm. | $55.12 \pm 0.17$ | $56.93 \pm 0.26$ | $\mathbf{57.64 \pm 0.36}$ | $57.47 \pm 0.66$ | $57.13 \pm 0.34$ |
| 97% | LTH | $51.10 \pm 0.34$ | $56.14 \pm 0.56$ | $56.92 \pm 0.25$ | $56.94 \pm 0.13$ | $56.93 \pm 0.06$ |
| | naive | $49.70 \pm 0.64$ | $49.60 \pm 0.25$ | $49.49 \pm 0.32$ | $49.16 \pm 0.21$ | $47.70 \pm 0.83$ |
| | perm. | $50.34 \pm 0.21$ | $51.55 \pm 0.69$ | $51.88 \pm 1.08$ | $\mathbf{52.64 \pm 0.34}$ | $50.96 \pm 1.15$ |

*Table 12.* **ResNet20$\times$\{4\}/CIFAR-100**. Results using the ResNet20$\times$\{4\} trained on CIFAR-100, from a rewind point $k$, using various methods of sparse training with sparsity $S$. LTH trains within the original dense/pruned solution basin, while naive/permuted train from a new random initialization. Note this table is the same setting as Table 11 except $w=4$.

| $S$ | Method | Rewind Epoch $k$ | | | | |
| --- | --- | --- | --- | --- | --- | --- |
| | | $k$=0 | 10 | 25 | 50 | 100 |
| 80% | LTH | $74.46 \pm 0.12$ | $77.57 \pm 0.06$ | $77.35 \pm 0.31$ | $77.75 \pm 0.26$ | $77.64 \pm 0.14$ |
| | naive | $73.30 \pm 0.08$ | $74.10 \pm 0.12$ | $74.98 \pm 0.17$ | $75.21 \pm 0.12$ | $75.20 \pm 0.16$ |
| | perm. | $73.68 \pm 0.09$ | $75.24 \pm 0.31$ | $75.74 \pm 0.41$ | $76.12 \pm 0.37$ | $\mathbf{76.19 \pm 0.39}$ |
| 90% | LTH | $72.54 \pm 0.57$ | $76.56 \pm 0.11$ | $76.56 \pm 0.32$ | $76.80 \pm 0.34$ | $76.80 \pm 0.21$ |
| | naive | $71.97 \pm 0.30$ | $72.56 \pm 0.22$ | $72.89 \pm 0.27$ | $72.59 \pm 0.15$ | $72.54 \pm 0.33$ |
| | perm. | $72.18 \pm 0.23$ | $74.17 \pm 0.35$ | $74.21 \pm 0.23$ | $74.45 \pm 0.27$ | $\mathbf{74.89 \pm 0.47}$ |
| 95% | LTH | $71.16 \pm 0.23$ | $75.41 \pm 0.18$ | $75.53 \pm 0.11$ | $75.68 \pm 0.17$ | $75.76 \pm 0.17$ |
| | naive | $70.17 \pm 0.47$ | $70.95 \pm 0.50$ | $70.90 \pm 0.18$ | $71.21 \pm 0.26$ | $69.95 \pm 0.42$ |
| | perm. | $70.41 \pm 0.07$ | $72.70 \pm 0.21$ | $72.92 \pm 0.39$ | $\mathbf{73.65 \pm 0.28}$ | $73.41 \pm 0.18$ |
| 97% | LTH | $69.06 \pm 0.03$ | $74.00 \pm 0.39$ | $74.08 \pm 0.37$ | $74.18 \pm 0.18$ | $74.29 \pm 0.31$ |
| | naive | $68.40 \pm 0.21$ | $69.26 \pm 0.19$ | $69.06 \pm 0.11$ | $68.67 \pm 0.47$ | $68.42 \pm 0.78$ |
| | perm. | $69.08 \pm 0.22$ | $71.41 \pm 0.54$ | $71.49 \pm 0.32$ | $71.92 \pm 0.17$ | $\mathbf{72.20 \pm 0.08}$ |

*Table 13.* **ResNet20×{8}/CIFAR-100**. Results using the ResNet20×{8} trained on CIFAR-100, from a rewind point $k$, using various methods of sparse training with sparsity $S$. LTH trains within the original dense/pruned solution basin, while naive/permuted train from a new random initialization. Note this table is the same setting as Table 11 except $w=8$.

| | | Rewind Epoch $k$ | | | | |
|---|---|---|---|---|---|---|
| $S$ | Method | $k=0$ | 10 | 25 | 50 | 100 |
| 80% | LTH | $78.09 \pm 0.28$ | $80.63 \pm 0.32$ | $80.83 \pm 0.39$ | $80.92 \pm 0.06$ | $80.66 \pm 0.34$ |
| | naive | $76.86 \pm 0.17$ | $77.47 \pm 0.35$ | $78.20 \pm 0.61$ | $78.65 \pm 0.33$ | $78.74 \pm 0.39$ |
| | perm. | $77.34 \pm 0.26$ | $78.82 \pm 0.34$ | $79.20 \pm 0.16$ | $\mathbf{79.55 \pm 0.38}$ | $79.54 \pm 0.39$ |
| 90% | LTH | $76.47 \pm 0.43$ | $80.02 \pm 0.07$ | $80.10 \pm 0.13$ | $79.98 \pm 0.33$ | $79.98 \pm 0.20$ |
| | naive | $75.68 \pm 0.23$ | $76.36 \pm 0.21$ | $76.80 \pm 0.14$ | $77.27 \pm 0.12$ | $76.55 \pm 0.49$ |
| | perm. | $76.17 \pm 0.26$ | $77.99 \pm 0.17$ | $78.22 \pm 0.15$ | $78.62 \pm 0.19$ | $\mathbf{78.82 \pm 0.17}$ |
| 95% | LTH | $75.38 \pm 0.02$ | $79.42 \pm 0.06$ | $79.24 \pm 0.19$ | $79.35 \pm 0.06$ | $79.29 \pm 0.13$ |
| | naive | $74.78 \pm 0.15$ | $75.48 \pm 0.18$ | $75.53 \pm 0.15$ | $75.27 \pm 0.15$ | $74.38 \pm 0.65$ |
| | perm. | $75.07 \pm 0.14$ | $76.97 \pm 0.46$ | $77.80 \pm 0.14$ | $77.74 \pm 0.51$ | $\mathbf{78.04 \pm 0.42}$ |
| 97% | LTH | $73.97 \pm 0.21$ | $78.63 \pm 0.25$ | $78.65 \pm 0.50$ | $78.74 \pm 0.49$ | $78.47 \pm 0.16$ |
| | naive | $73.13 \pm 0.26$ | $73.73 \pm 0.12$ | $73.76 \pm 0.27$ | $73.26 \pm 0.07$ | $72.79 \pm 0.46$ |
| | perm. | $73.81 \pm 0.67$ | $76.29 \pm 0.14$ | $76.38 \pm 0.57$ | $76.57 \pm 0.29$ | $\mathbf{76.79 \pm 0.76}$ |

*Table 14.* **ResNet20×{16}/CIFAR-100**. Results using the ResNet20×{16} trained on CIFAR-100, from a rewind point $k$, using various methods of sparse training with sparsity $S$. LTH trains within the original dense/pruned solution basin, while naive/permuted train from a new random initialization. Note this table is the same setting as Table 11 except $w=16$.

| | | Rewind Epoch $k$ | | | | |
|---|---|---|---|---|---|---|
| $S$ | Method | $k=0$ | 10 | 25 | 50 | 100 |
| 80% | LTH | $80.21 \pm 0.18$ | $82.32 \pm 0.34$ | $82.40 \pm 0.26$ | $82.48 \pm 0.38$ | $82.16 \pm 0.30$ |
| | naive | $79.31 \pm 0.06$ | $79.50 \pm 0.09$ | $80.24 \pm 0.17$ | $81.02 \pm 0.11$ | $81.01 \pm 0.07$ |
| | perm. | $79.35 \pm 0.11$ | $80.44 \pm 0.40$ | $81.15 \pm 0.48$ | $81.57 \pm 0.38$ | $\mathbf{81.81 \pm 0.21}$ |
| 90% | LTH | $79.31 \pm 0.16$ | $82.26 \pm 0.18$ | $82.14 \pm 0.08$ | $81.95 \pm 0.03$ | $82.11 \pm 0.12$ |
| | naive | $78.78 \pm 0.37$ | $79.26 \pm 0.11$ | $79.42 \pm 0.51$ | $79.56 \pm 0.26$ | $79.57 \pm 0.13$ |
| | perm. | $79.20 \pm 0.09$ | $80.49 \pm 0.32$ | $80.59 \pm 0.15$ | $81.12 \pm 0.05$ | $\mathbf{81.24 \pm 0.09}$ |
| 95% | LTH | $78.32 \pm 0.34$ | $81.57 \pm 0.09$ | $81.57 \pm 0.32$ | $81.47 \pm 0.25$ | $81.63 \pm 0.07$ |
| | naive | $78.01 \pm 0.02$ | $78.53 \pm 0.10$ | $78.45 \pm 0.21$ | $78.38 \pm 0.43$ | $77.49 \pm 0.06$ |
| | perm. | $78.25 \pm 0.20$ | $79.76 \pm 0.20$ | $\mathbf{80.50 \pm 0.04}$ | $80.47 \pm 0.08$ | $80.25 \pm 0.21$ |
| 97% | LTH | $77.49 \pm 0.27$ | $81.07 \pm 0.07$ | $81.06 \pm 0.11$ | $81.11 \pm 0.18$ | $81.14 \pm 0.32$ |
| | naive | $76.46 \pm 0.44$ | $76.71 \pm 0.41$ | $77.19 \pm 0.09$ | $76.93 \pm 0.36$ | $75.53 \pm 0.40$ |
| | perm. | $77.04 \pm 0.38$ | $79.14 \pm 0.17$ | $79.30 \pm 0.21$ | $79.62 \pm 0.14$ | $\mathbf{79.63 \pm 0.06}$ |

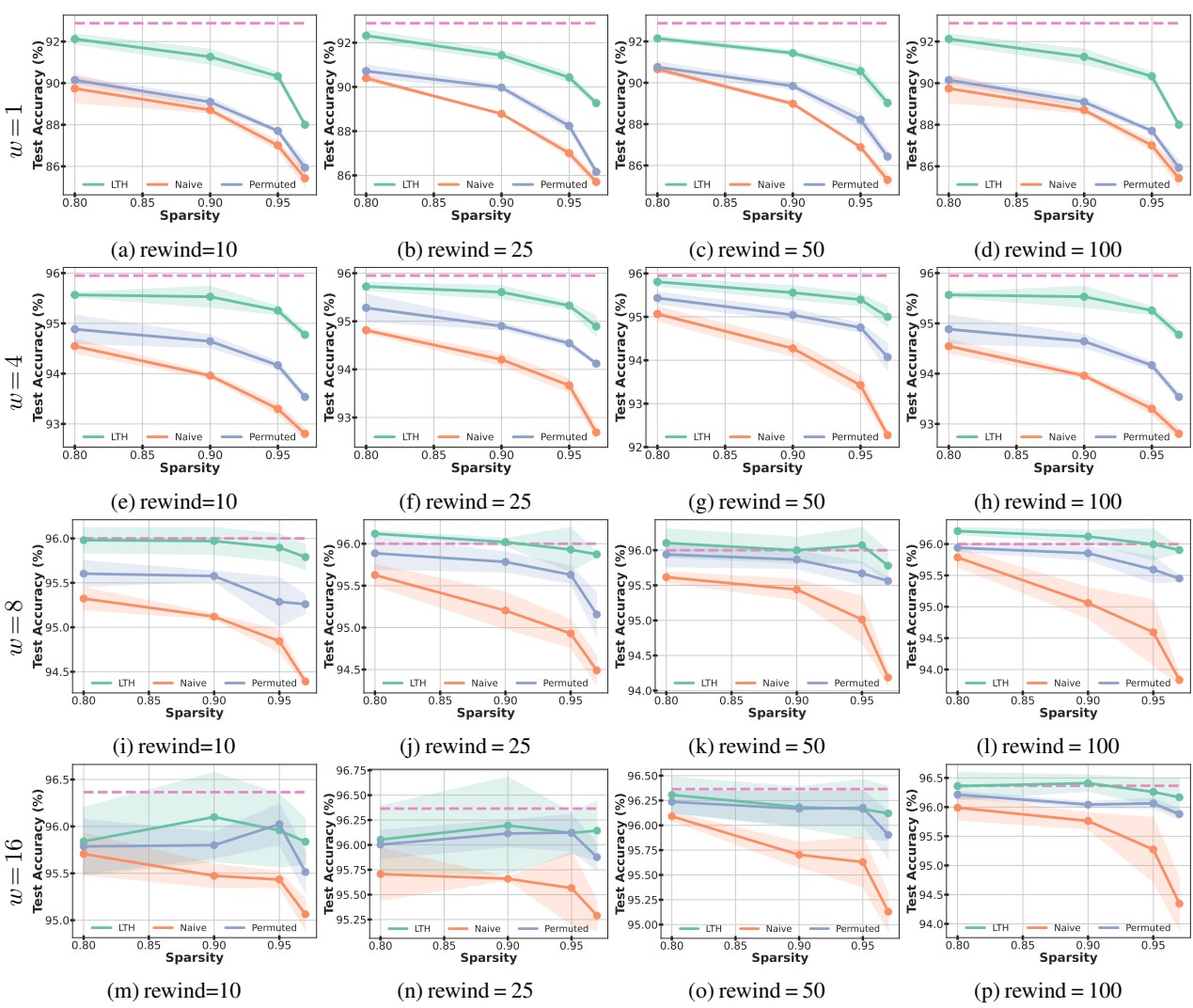

*Figure 9.* **Accuracy vs sparsity trend for ResNet20**$\times\{w\}$**/CIFAR-10**. As the width increases, the gap between permuted and naive solutions increases, showing permuted masks help with sparse training. With increased width, we observe a more significant gap seen throughout Figures 9d, 9h, 9l and 9p and the permuted solution approaches the LTH solution. The dashed (**- -**) line shows the dense model accuracy.

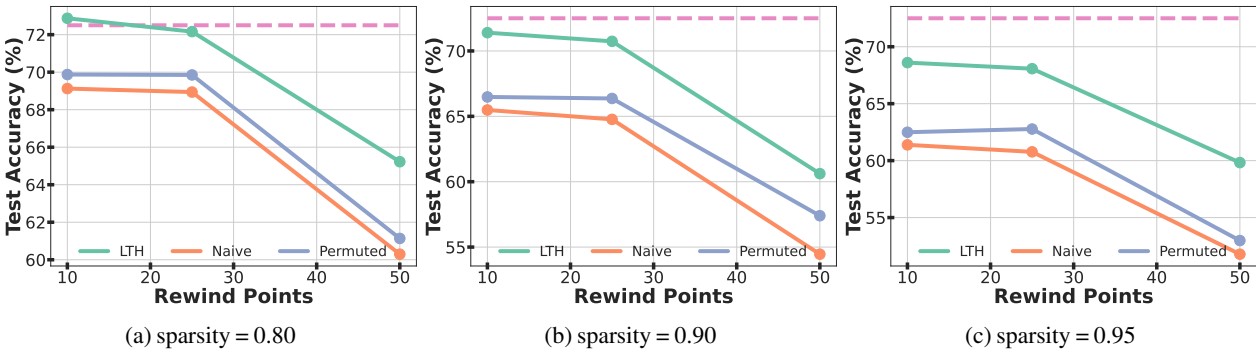

*Figure 10.* **Accuracy vs sparsity trend for ResNet20×{w}/CIFAR-100**. Similar to the phenomenon seen in Figure 9, with higher width, the gap between permuted and naive solutions increases. As seen in Figures 10d, 10h, 10l and 10p and the permuted solution approaches the LTH solution. The dashed (- -) line shows the dense model accuracy.

*Figure 11.* **ResNet50×{1}/ImageNet**. Top-1 test accuracy vs rewinds points of sparse network solutions at various sparsity levels. We observe the permuted solution consistently performing better than the naive solution for all sparsities. The dashed (- -) line shows the dense model accuracy.

*Table 15.* **ResNet20×{1}/CIFAR-10**. Results using the ResNet20×{1} trained on CIFAR-10, from a rewind point $k = 20$, using various methods of sparse training with sparsity $S$. The LTH and naive methods remain fixed as they are independent of matching. For the permuted method, the permutation, $\pi$, is obtained by matching a fully trained dense model at $t = T$ ($T = 200$) with another model at an early point in training at $t = i$, where $i \in \{5, 20, 50, 100\}$.

| | | Early Matching Point $t$ | | | | |
|---|---|---|---|---|---|---|
| $S$ | Method | $t = 5$ | 20 | 50 | 100 | 200 |
| 80% | LTH | | | $92.25 \pm 0.14$ | | |
| | naive | | | $90.13 \pm 0.11$ | | |
| | perm. | $90.49 \pm 0.37$ | $90.34 \pm 0.63$ | $90.42 \pm 0.29$ | $90.42 \pm 0.25$ | $90.68 \pm 0.18$ |
| 90% | LTH | | | $91.18 \pm 0.27$ | | |
| | naive | | | $88.83 \pm 0.27$ | | |
| | perm. | $89.16 \pm 0.51$ | $89.23 \pm 0.59$ | $89.39 \pm 0.69$ | $89.31 \pm 0.60$ | $89.50 \pm 0.27$ |
| 95% | LTH | | | $90.58 \pm 0.26$ | | |
| | naive | | | $87.31 \pm 0.36$ | | |
| | perm. | $87.37 \pm 0.33$ | $87.68 \pm 0.77$ | $87.43 \pm 1.00$ | $87.54 \pm 0.43$ | $88.29 \pm 0.52$ |
| 97% | LTH | | | $89.21 \pm 0.23$ | | |
| | naive | | | $85.36 \pm 0.14$ | | |
| | perm. | $85.77 \pm 0.44$ | $85.93 \pm 0.94$ | $86.09 \pm 0.51$ | $85.88 \pm 0.47$ | $86.12 \pm 0.27$ |

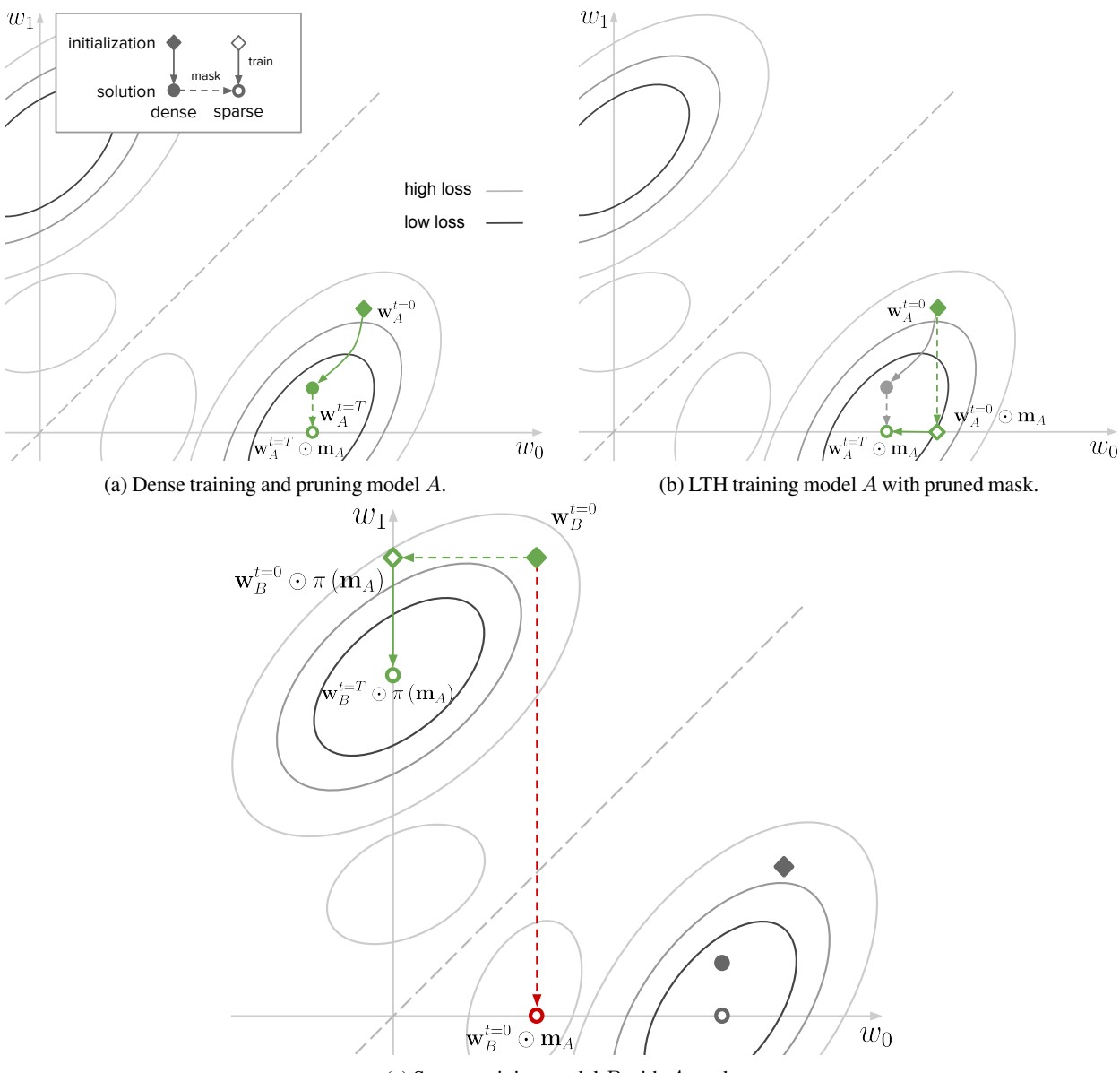

(a) Dense training and pruning model $A$.

(b) LTH training model $A$ with pruned mask.

(c) Sparse training model $B$ with $A$ mask.

*Figure 12.* **Weight Symmetry and the Sparse Training Problem (Full Figure)**. A model with a single layer and only two parameters, $\mathbf{w} = (w_0, w_1)$, operating on a single input $x_0$ has the weight symmetry in the 2D loss landscape as illustrated above. In (a) the original dense model, $\mathbf{w}_A$, is trained from a random dense initialization, $\mathbf{w}_A^{t=0}$ to a dense solution, $\mathbf{w}_A^{t=T}$, which is then pruned using weight magnitude resulting in the mask $\mathbf{m}_A = (1,0)$. In (b) we re-use the init. $\mathbf{w}_A^{t=0}$, to train model $A$ with the pruned mask from (a), $\mathbf{m}_A$, as in the LTH. In (c), naively using the same mask to train a model, B, from a different random initialization will likely result in the initialization being far from a good solution. Permuting the mask to match the (symmetric) basin in which the new initialization is in will enable sparse training.

