# OpenReview forum: "Sparse Training from Random Initialization: Aligning Lottery Ticket Masks using Weight Symmetry"
_ICML.cc/2025/Conference — ICML 2025 poster_

### Official Review · Reviewer_yfeq · 2025-02-20

**Overall Recommendation:** 3

**Summary:**

The authors investigate the problem of LTH masks not being compatible with random initializations. They find that by aligning the loss basins via a matching permutation, an LTH mask can be used with a random initialization, not associated with the mask.
This work shows that LTH masks can be reused with random initializations via permutation matching (to some extent).

**Claims And Evidence:**

The claims made in the paper are supported empirically via experiments on CIFAR10, CIFAR100 and also the ImageNet datasets.

**Essential References Not Discussed:**

The authors discuss the relevant literature.

**Ethical Review Concerns:**

No ethical concerns.

**Experimental Designs Or Analyses:**

The conclusions drawn from the ensemble experiments are a bit unclear to me (see questions below).

**Methods And Evaluation Criteria:**

Yes, the evaluation methods (generalization performance) and criteria (image classification datasets) are appropriate for this problem setting.

**Other Comments Or Suggestions:**

I appreciate the authors’ insight on using permutations to make LTH masks flexible and reusable with random initializations; the improvements in performance seem limited to me. I believe the paper will benefit from an explanation of why the performance gains are limited (is it due to the limitation of permutation matching or due to the fact that the hypothesis of misaligned basins is not enough). Hence I lean towards a reject at the moment, but I am happy to increase my score after a discussion with the authors during the rebuttal.

**Other Strengths And Weaknesses:**

Strengths
1. Simply permuting the mask can help match the LTH mask to any random initialization.
2. This improves the performance of the random initialization with the random mask.

Weaknesses (and questions)
1. It seems that the permutation can only be identified by training two dense networks, is this necessary or can this be done on a sparse network as well?
2. The hypothesis that the difference between a random initialization and the LTH is due to the misalignment of the loss basin seems limited. Because via permutation matching, the random init with the LTH mask is only slightly better than the naive baseline. Significant improvements are only seen on wider networks, even in the case of CIFAR10 and CIFAR100. This suggests that there might be more than a misalignment of basins or the permutation matching is not good enough, maybe the authors could investigate this?
3. What is the difference between IMP and LTH in Table 1, is IMP trained over 5 different seeds and LTH uses the same mask and init over 5 different runs? This will naturally have a smaller ensemble effect since the only randomness is the stochastic noise compared to different initializations in the case of IMP. The permutations however, have different random initializations, which would help ensembling. In spite of this, the permuted solutions are similar or worse than IMP. Can the authors explain the experimental setup in more detail to highlight the differences between the permuted ensemble and the LTH ensemble. Otherwise, the functional diversity of permutations is unclear to me.
4. Low width networks observe a smaller improvement with permutation, is there a reason for this? Is there a tradeoff between the alignment of the mask and initialization (via permutation) and the amount of overparameterization (width).

**Questions For Authors:**

See weaknesses above.

**Relation To Broader Scientific Literature:**

This paper addresses the problem of making LTH masks useful for other random initializations, which is a step in the direction to understand how sparse networks can be trained from scratch. This is an active area of research.

**Theoretical Claims:**

There are no theoretical claims in this paper.

---

> ### Author Rebuttal · Authors · 2025-03-28
>
> We sincerely appreciate the detailed feedback provided; we provide more details below:
>
> 1.
>
> In our work, we used 2 trained dense models to find permutation (perm) mapping, as primary aim of the paper was to understand why winning tickets don't work with different random init.
> One can also find perm mapping early in training, as noted by [Sharma, 2024]. We have added an additional experiment on early matching with CIFAR-10, which shows that models can be matched earlier in training, thus reducing computational cost of our method. (https://imgur.com/a/BgiE4W3)
>
> We don't think sparse models can be used for finding perm mapping as the mask projects the model into a different sub-space, thus cannot be matched with a perm.
>
> 2.
>
> Your observation is indeed correct! As noted in manuscript (L188), the perm matching algo uses a greedy approach and thus finds an approximate solution, i.e., the perm matching is not good enough (as you noted). As discussed in Sec 4.3, perm matching works better for wider models [2].
> This can be observed in the LMC plot (Fig 3.), where the loss barrier decreases when the model width is increased, showing that the perm matching finds a better solution as we increase the width.
> Our experiments in Sec 4.3 show that as we increase the model width, gap between LTH and permuted mask decreases, which suggests that the permuted solution will closely match LTH performance if given an accurate permutation mapping.
>
> 3.
>
> >What is the difference between IMP and LTH in Table 1? Is IMP trained over 5 different seeds and LTH uses the same mask and init over 5 different run
>
> Yes, IMP is trained independently over 5 different seeds with iterative pruning to obtain 5 different sparse/pruned solutions with different sparse masks/topologies ($M_0$, $M_1$, $M_2$, $M_3$, $M_4$, $M_5$).
> LTH ensemble is trained using the same mask ($M_0$) and init ($w_0$) over 5 different runs (with different data order). Random init $w_0$ defines the winning ticket for mask $M_0$.
> The permuted ensemble is trained using 5 different permutations ($\pi_1$, $\pi_2$, $\pi_3$, $\pi_4$, $\pi_5$) of the same mask ($M_0$) with five different random inits ($w_1$, $w_2$, $w_3$, $w_4$, $w_5$).
>
> >  The permutations however, have different random initializations. In spite of this, the permuted solutions are similar or worse than IMP.
>
> This is an interesting question! Our intuition is that in the case of IMP,  models are trained with different random initializations, and they discover different sparse topologies, which helps in learning more diverse solutions but is computationally expensive and not practical to create ensembles. The IMP baselines serve as an upper bound for diversity metrics as both training from different random initializations and learning different topologies introduce high randomness/stochasticity to learn very diverse solutions.
>
> LTH, as noted in prior work, does not learn diverse solutions as they are trained using the same mask and init but over different runs [1]. Thus, LTH is not suitable for generating ensembles.
>
> In contrast, our proposed method allows us to reuse the LTH mask with different random inits, introducing more source of randomness than the LTH baseline and thus improving the diversity of the solution. However, since we reuse the mask to train permuted ensembles, the diversity will be less than IMP (which uses both different init and sparse topologies) but good enough for making ensembles. This can be observed in Table 1., where the LTH ensemble does not improve the accuracy compared to a single model, while the permuted ensemble significantly improves the performance compared to a single model. It is worth noting that for CIFAR-100 datasets, the permuted ensemble (77.85%) surpasses the LTH ensemble (75.99%), demonstrating that the permuting mask can help train more diverse solutions with less computational cost incurred as compared to IMP.
>
> We will add more explanations and define LTH, IMP, and Permuted ensembles for an easier understanding.
>
> 4.
> > "Low width networks observe a smaller improvement with permutation"
>
> The perm matching algorithm doesn't work well with lower widths, which can be observed in the loss-barrier plots (high barrier --> poor perm matching). We still observe smaller but statistically significant improvements with the permuted mask, as discussed below. As we increase the width, perm matching becomes better, and the loss barrier decreases (fig 3.); our proposed method with permuted mask becomes closer to LTH.
>
> Even at width=1, we can see significant improvements at 97% sparsity:
>
> **CIFAR-10**: +1%
>
> **CIFAR-100**: +3.5%
>
> **Imagenet** (at 95% sparsity, width=1):  +2% (top-1)
>
> We have added more experiments in our reply to reviewer **oZBY**, which you may also find interesting.
> If you're satisfied with our explanation, we'd greatly appreciate you updating your score.
>
> [1] Evci et al, Gradient Flow in Sparse Neural Networks
>
> [2]. Ainsworth et al., Git Re-Basin

---

> > ### Comment · Reviewer_yfeq · 2025-04-02
> >
> > I thank the authors for the detailed response.
> >
> > **Permutations matching via sparse network**
> > From the additional experiments, it seems that the permutation can be identified already midway through dense training. Given that the subsequent sparse networks in IMP are linearly connected to this dense net, it should still be possible to find the permutation with the sparse network? Possibly at a higher sparsity this is harder. Or does the permutation matching also become worse for a sparser network?
> >
> > In order to clarify the reasons for the limited performance gains of the LTH with a random init, in spite of permutation matching, I would urge the authors to add the discussion regarding performance of the matching algorithm and the effect of width on the overall performance.
> >
> > The authors have addressed my questions sufficiently and I have updated my score accordingly.

---

> > > ### Author Response · Authors · 2025-04-03
> > >
> > > Thank you for taking the time to reply to our rebuttal and raising the score. We really appreciate it!
> > >
> > > **Matching via sparse network**:
> > > >subsequent sparse networks in IMP are linearly connected to this dense net, it should still be possible to find the permutation with the sparse network?
> > >
> > > It is indeed possible to match sparse networks found using the IMP, as shown by Sharma et al.. (https://imgur.com/a/mIpXNuv)
> > > However, obtaining the second sparse network with IMP would be computationally more expensive than early matching with a new dense model.
> > >
> > > > Possibly at a higher sparsity this is harder. Or does the permutation matching also become worse for a sparser network?
> > >
> > > Your intuition is indeed correct! The loss barrier increases as we increase the sparsity, indicating that it is difficult to find the permutation at higher sparsities.
> > >
> > > > In order to clarify the reasons for the limited performance gains of the LTH with a random init, in spite of permutation matching, I would urge the authors to add the discussion regarding the performance of the matching algorithm and the effect of width on the overall performance.
> > >
> > > We will surely add more discussion on the performance of weight/activation matching the width of the model.
> > >
> > > We hope we have answered all your questions; please let us know if you have more follow-up questions. We thank you for your valuable insights and for improving our paper.
> > >
> > > 1. Sharma et al., Simultaneous linear connectivity of neural networks modulo permutation

---

### Official Review · Reviewer_nWHn · 2025-03-07

**Overall Recommendation:** 2

**Summary:**

This paper hypothesizes that Iterative Magnitude Pruning (IMP) fails to generalize its sparse mask to other random initialization because the basin in which other random initialization resides does not match the basin constructed by the IMP sparse mask.
To address this, the authors propose to permute the IMP mask to align with the basin of other random initialization.
The authors evaluate the proposed method on CIFAR10/100 and ImageNet datasets with VGGNet and ResNet.

## Update after rebuttal

The authors' response did not effectively address the raised concerns, particularly about the gap between the authors' claim and the proposed method. Thus, the reviewer still hesitates to accept this work because the current version is confusing and unconvincing. However, the reviewer has decided to raise the original rating to 'weak reject', expecting the authors to accept the suggestions detailed in the comments.

**Claims And Evidence:**

The authors claim that the IMP sparse mask may not match the basin in which other random initialization resides.
The evidence for this claim is just the analysis in Figure 1 with the assumption of a single layer with two parameters case.

However, previous work [1] discovered that an initialized model often exhibits insufficient stability to SGD noise, meaning that it is not in a basin of attraction.
Thus, the assumption that an initialized model is supposed to be in a specific basin may not be true.
Also, it is difficult to agree that a sparse mask determines a specific basin of attraction according to the findings in [1].

Therefore, the reviewer is not convinced by the claims of this paper.

[1] Frankle et al, "Linear mode connectivity and the lottery ticket hypothesis", ICML2020.

**Essential References Not Discussed:**

None.

**Experimental Designs Or Analyses:**

The authors show the experimental results with varying widths of networks, showing that the proposed method is more effective with large widths.
This claim seems to be contrary to the goal of network sparsification.
If the proposed sparsification method is effective with larger models, it would be meaningless.

**Methods And Evaluation Criteria:**

The authors propose permutation matching to match the basin of an initialized model and a sparse model.
However, this method is just adopted from previous works without any significant modification.
Thus, this paper seems to lack technical novelty.

For evaluation, the authors use CIFAR10/100 and ImageNet datasets, which are commonly used benchmark datasets.

**Other Comments Or Suggestions:**

In the main manuscript, the authors use 'LTH mask' repeatedly.
However, it is difficult to find the details of which methods the authors use in the main manuscript and I managed to find the details in the Appendix.
Similarly, in experimental sections, It is difficult to find what 'Naive' refers to.
Overall, a clearer and more reader-friendly presentation seems to be required.

**Other Strengths And Weaknesses:**

All are mentioned in other sections.

**Questions For Authors:**

None.

**Relation To Broader Scientific Literature:**

The contribution of this paper seems to be limited.
The authors' assumption is not convincing and the experimental results are also not significant.

**Theoretical Claims:**

There is no theoretical claim.

---

> ### Author Rebuttal · Authors · 2025-03-28
>
> >The authors claim that the IMP sparse mask may not match the basin in which other random init resides. evidence for this claim is just analysis in Figure 1 with assumption of a single layer with two parameters case.
>
> We validate our hypothesis through comprehensive experiments conducted across multiple datasets and model architectures. Our findings are substantiated by similar observation in [3], which demonstrated that IMP sparse masks only work when dense and sparse models are within the same loss basin and linearly mode-connected.
>
> Our hypothesis is intuitive: by matching loss basins of two models with different initializations, we can permute and reuse the IMP mask from a rewind point.
> Our experimental results provide empirical evidence supporting this claim, showing consistent improvement across different model arch and datasets.
>
> > Thus, the assumption that an initialized model is supposed to be in a specific basin may not be true.
>
> We agree that initialization alone does not determine the basin. However, **we do not claim this anywhere in the manuscript**. Models trained from the same initialization can end up in different basins [1]. That’s why using a rewind point for Lottery Tickets is necessary, and we use a rewind point with our method as well. We will add a note on this in final version of manuscript to make it clearer.
>
> Our method builds upon the fact that models trained using LTH mask (with rewind points) always land in the same basin, as the authors observed in [2]. Our key claim is that the winning mask can be used with a rewind point obtained from different init, provided we account for weight symmetry to align the basins. We empirically demonstrate this through extensive experiments across multiple datasets and model architectures.
>
> > If the proposed sparsification method is effective with larger models, it would be meaningless.
>
> We *respectfully* disagree with this statement. Even with width=1,  we can observe statistically significant improvement for different datasets as shown in Tables 5, 9, 10, 11.
>
> * **CIFAR-10** (at 97% sparsity, width=1) - Improvement of **1%**
> * **CIFAR-100** (at 97% sparsity, width=1) - Improvement of **3.5%**
> * **Imagenet** (at 95% sparsity, width=1) - Improvement of **2%** (top-1)
>
> These improvements are statistically significant and demonstrate the efficacy of our method. In the manuscripts, we added experiments with varying widths to get more insights about the accuracy of permutation matching and show that once permutation matching gets better on increasing the width, the gap between LTH and permuted mask (our method) reduces. This experiment provides more insight into the role of permutation matching for our proposed method. You can find more details in **Sec 4.3**.
>
> We would also like to highlight **our work aims at better understanding of lottery tickets and winning masks, not just improving the accuracy**.  As noted by reviewer **oZBY**,  our work "provides novel insights into the relationship between winning tickets and their original dense networks.” We believe our findings will be useful for the sparse training research community.
>
> > In the main manuscript, the authors use 'LTH mask' repeatedly. However, it is difficult to find the details of which methods the authors use in the main manuscript and I managed to find the details in the Appendix. Similarly, in experimental sections, It is difficult to find what 'Naive' refers to. Overall, a clearer and more reader-friendly presentation seems to be required.
>
> We appreciate the suggestion; we will add a separate paragraph to define naive, LTH and permuted masks for an easier understanding.
>
> However, we would like to point out that we have defined the *LTH*, *naive*, and *permuted mask* at multiple places in the manuscript (**first one at L94-96**; see more below). Moreover, **Figure 2** in the manuscript explains the differences between LTH, naive and permuted baselines.
>
> * **Line 94-96**: "Permuting the LTH sparse mask to align with
> the new random initialization improves the performance
> of the trained model (**permuted**), compared to the model
> trained without permuting the sparse mask (**naive**)."
>
> * **Line 190-192**: "We denote training with the permuted mask, $\pi(\textbf{m}_A)$
> as **permuted** and with the non-permuted mask, $\textbf{m}_A$ as **naive**"
>
> * **Line 212-214**: "To evaluate the transferability of the permuted LTH mask we train, a different random initialization $\textbf{w}_B^{t=0}$, the LTH sparse mask $\textbf{m}_A$ and permuted LTH mask $\pi(\textbf{m}_A)$, which we denote the **naive** and **permuted** solution
> respectively."
>
> If you're satisfied with our reply, we'd appreciate if you can update your score.
>
> [1]. Frankle et al., Linear mode connectivity and the lottery ticket hypothesis
>
> [2] Evci et al, Gradient Flow in Sparse Neural Networks and How Lottery Tickets Win
>
> [3]. Paul et al., Unmasking the Lottery Ticket Hypothesis: What's Encoded in a Winning Ticket's Mask

---

> > ### Comment · Reviewer_nWHn · 2025-04-05
> >
> > Thanks for the authors' detailed responses.
> >
> > The responses resolve several concerns about model widths and the unclear definition of terms.
> >
> > However, the other concerns remain unsolved.
> >
> > L87-L90 (left column) and L153-L157 (right column) say the misalignment between the basin corresponding to LTH mask and **the basin of new random initialization**.
> > These sentences likely lead a reader to believe that the authors claim there exists an expected basin corresponding to a pruning mask and a random initialization, and the authors want to make any random initialization a winning ticket using a single LTH mask.
> > What these sentences mean is quite different from the authors' claim that "the winning mask can be used with a rewind point obtained from a different initialization.".
> > That's why the reviewer is confused by the authors' claim.
> > The reviewer finds that the current version is likely to mislead readers and that clearer and more accurate expressions are needed.
> >
> > Moreover, *weight rewinding* [1] was proposed not to find a winning ticket but to better understand why the original IMP fails.
> > Thus, improving *weight rewinding* by making a single LTH mask generalizable to any rewind point, rather than any initialization, does not seem to offer a significant contribution.
> > Also, given the concern about technical novelty—which the authors did not address—it is difficult to support the acceptance of this paper to ICML.
> >
> >
> >
> > [1]. Frankle et al., Linear mode connectivity and the lottery ticket hypothesis

---

> > > ### Author Response · Authors · 2025-04-06
> > >
> > > We think there is some misunderstanding in this discussion about original LTH work by Frankle et al., the paper you cited [3], and our work. We briefly review the original LTH papers and the paper you cited to set the motivation for our contribution and to highlight how our work is important for better understanding of LTH and sparse training, as noted by other reviewers.
> > >
> > > **History of LTH**
> > >
> > >      1. LTH was introduced by Frankle et al. [1], who hypothesized the existence of winning tickets at initialization. However, in this paper, they only experimented with small models and datasets and found that LTH from random init doesn’t work for larger models.
> > >
> > >      2. Quoting directly from paper: "We only consider vision-centric classification task on smaller dataset (MNIST, CIFAR10). We do not investigate larger dataset."
> > >
> > >     3. Follow-up paper [2] from Frankle et al. proposed that for LTH to work on larger models, we need to apply the mask at the rewind point (not at init) and linked this to SGD instability: "In this paper, we demonstrate that there exist subnetworks of deeper networks at early points in training"  (page 2).
> > >
> > >     4. Frankle et al. added more analysis in subsequent version of the paper—which you cited [3]—and studied SGD stability by analyzing linear mode connectivity:
> > >      "We introduce instability analysis to determine whether the outcome of optimizing a neural network is stable to SGD noise, and we suggest linear mode connectivity for making this determination."
> > >      "We show that IMP subnetworks become stable and matching when set to weights from early in training, making it possible to extend the lottery ticket observations to larger scales."
> > >
> > > **Summary**: The LTH mask only works at the rewind point and not at the init. The paper you cited suggests this is due to the stability of SGD noise.
> > >
> > > **Limitations of LTH**: The LTH mask can't be used with a new init, making it impractical. It is not clear why the LTH mask can't be reused to a new random init. Ensembles trained with LTH do not work well because LTH learns similar functions [4].
> > >
> > > **Our contribution**: We show how we can use the LTH mask with a new random init by leveraging weight symmetry.
> > > We also show that our method can help improve diversity of sparse models, which help in improving ensemble significantly as reviewer oZBY noted.
> > >
> > > **We now address your comments below**:
> > >
> > > >  Moreover, *weight rewinding* [1] was proposed not to find winning ticket but to better understand why the original IMP fails.
> > >
> > > This is incorrect, weight rewinding was proposed to allow LTH to work for larger models/datasets as cited above.  The paper you cited tried to understand why the mask from IMP cannot be applied at init directly and why it only works with a bit of pre-training, aka, at the rewind point.
> > >
> > > > Thus, improving *weight rewinding* by making a single LTH mask generalizable to any rewind point, rather than any init, does not seem to offer a significant contribution
> > >
> > > There is again some misunderstanding here. Our work precisely aims to make the LTH mask more generalizable to new random inits, as noted by other reviewers. We do this over a range of rewind points to maintain existing LTH methodology. We take a new random init and do a little training up to rewind point. Our method allows us to reuse the same LTH mask with arbitrary random inits. As you said, making the LTH mask generalizable is important, and **our work does precisely that**. Other reviewers have the same understanding of our contribution:
> > >
> > > Reviewer  **oZBY** :
> > > "The authors build upon those findings and show that **winning ticket masks can be reused for different weight initialization**."
> > >
> > > Reviewer **yfeq**:
> > > "I appreciate the authors' insight on **using permutations to make LTH masks flexible and reusable with random initialization.**"
> > >
> > > "Strengths:  Simply **permuting the mask can help match the LTH mask to any random initialization**. This improves the performance of random initialization with the random mask."
> > >
> > > > Technical novelty:
> > >
> > > Our work provides novel insight about why the LTH mask does not generalize to new init from weight symmetry perspective. As reviewer **oZBY** noted:
> > >
> > > "The paper provides novel insights into the relationship between winning tickets and their original dense networks."
> > >
> > > "This paper addresses the problem of making LTH masks useful for other random initializations, which is a step in the direction to understand how sparse networks can be trained from scratch. This is an active area of research."
> > >
> > > We thank you for the feedback; we’ll add more explanation to make it easier for readers to understand our work. We hope we’ve clarified your confusion about our contribution, and we’d greatly appreciate it if you could consider updating the score.
> > >
> > > [1]. The LTH: Finding Sparse, Trainable Neural Network
> > >
> > > [2] Stabilizing Lottery Ticket Hypothesis
> > >
> > > [3] Linear Mode Connectivity and the LTH
> > >
> > > [4]. Evci et al., Gradient Flow in Sparse Neural Network

---

### Official Review · Reviewer_oZBY · 2025-03-10

**Overall Recommendation:** 4

**Summary:**

The paper studies the property of winning tickets, where the combination of the sparse mask and initial weight values determines eventual generalization performance. When these networks are trained using the same sparse mask but different weight initializations, their performance deteriorates. This is a well known test for whether a sparse network at initialization is a winning ticket or not.

The authors hypothesize that this is due to the mismatch of the basins between different weight initialization and the winning ticket mask. This hypothesis was motivated by the recent discovery that dense networks trained from different random initializations find solutions within the same loss basin modulo permutation. The authors then test the hypothesis by finding a permutation function between two trained dense networks, and show that the permuted winning ticket mask corresponding to the first network can be applied to the second network at a rewind point with better performance as compared to naive application of the mask.

The analysis conducted demonstrates that sparse networks found in each iteration of the IMP are linearly connected in the loss landscape to the dense network when the variance collapse is accounted for. This allows for finding a permutation function after the dense networks are fully trained and not when both are fully pruned. The authors validate the hypothesis on a range of networks and datasets. The results show that the performance degradation of sparse networks trained with the permuted masks decreases with the width of the networks. Finally, it is also shown that the diversity of the solutions discovered is greater than the winning tickets.

**Claims And Evidence:**

The claims regarding misalignment between weight initialization-based optimization basins and winning ticket masks are well-supported by empirical evidence. The paper provides novel insights into the relationship between winning tickets and their original dense networks.

**Essential References Not Discussed:**

The paper is quite extensive in its discussions of relevant literature and it is one of the strengths of the paper.

**Experimental Designs Or Analyses:**

The experiments are very well set-up with clear conclusions. However, there is a lack of clarity regarding the iterative magnitude pruning process used to identify the winning ticket mask. In the appendix the authors specify the use of IMP-FT which does not include a weight rewinding step.

**Methods And Evaluation Criteria:**

The presented methods and evaluation criteria are well-suited for testing the hypothesis.

**Other Comments Or Suggestions:**

Please see the Questions section below and Weaknesses section above.

**Other Strengths And Weaknesses:**

Strengths:

* The paper is very well written, the arguments made and evidence provided is very convincing. Please see the response to previous sections

Weaknesses:

* The contributions, while valuable, are primarily extensions of closely related previous work.

* Permuted masks are obtained through a function that is optimized between two trained dense networks and is computationally expensive. The authors acknowledge this, however, a natural question that arises is whether the permutation function can be obtained early in training eliminating the requirement for fully training the new model ?

* To what degree the permuted sparse masks can be reused is unclear. For instance can those masks be reused across datasets ?

**Questions For Authors:**

* Can the randomly initialized neural network be permuted instead of the mask to obtain the same results, given that a permutation is found that transforms model B to match model A ?
* Can the authors clarify what they mean by rewind points? Is it simply the training iteration at which the masks were applied to various dense networks ? Is it also the point at which the weights were rewinded during IMP while obtaining the original mask (m_A) resulting in different masks for different rewind points ?
* In Figure 4, is the IMP used along with weight rewinding or is IMP-FT used ? If IMP-FT was used then does the linear mode connectivity (accounting for variance collapse) exist between the dense network and the sequential winning tickets of lower density?
* The results show a clear trend as the width of the networks increases. The authors show that this is due to better LMC. Can this be simply because the permutation matching is better because each unit has a larger pool of units to match from ? Additionally, the variance in the unit activations can be studied, larger width may result in lower variance and better match.

## Update after rebuttal

I thank the authors for their response and for generating the requested results. I believe the paper makes several significant contributions, I have improved the rating. However, I request the authors to make the requested changes in the next version of the paper.

**Relation To Broader Scientific Literature:**

The key contribution of the work is showing that winning ticket masks can be used for networks with different weight initialization provided a suitable permutation function is available. The results provide insights into winning ticket sparse masks and the correlation between the specific weights at initialization. This has been previously used as a test for other pruning at initialization methods [Lee2019, Wang2020, Tanaka2020], without a degradation in performance.

The idea that the entire SGD trajectory and the sparse networks obtained can be aligned for two networks via the same permutation has been explored previously [Sharma2024]. The authors build upon those findings and show that winning ticket masks can be reused for different weight initializations.

**Theoretical Claims:**

NA

---

> ### Author Rebuttal · Authors · 2025-03-29
>
> Thank you for the detailed feedback and helpful insights. We have added new experiments based on your insights/questions and added our response below:
>
> > a natural question that arises is whether the permutation function can be obtained early in training eliminating the requirement for fully training the new model?
>
> In our work, we used 2 trained dense models to find the permutation mapping, but we can also find the perm mapping early in the training, as noted by [Sharma, 2024]. We have added an additional experiment on early matching with the CIFAR-10 dataset, which shows that models can be matched earlier in the training, thus reducing the computational cost of our method.
> (https://imgur.com/a/BgiE4W3)
>
> >  For instance can those masks be reused across datasets ?
>
> We conducted an additional experiment where we obtained a mask for the CIFAR-10 dataset and reused the mask with a new init on SVHN datasets. Permuted mask outperforms unpermuted mask (naive) at all levels of sparsities. Thank you for this interesting direction, we will add more experiments in the final version of the paper. (https://imgur.com/a/UDiMQHs)
>
> > Can the randomly initialized neural network be permuted
>
> Yes, indeed, we can also apply the permutation, $\pi$, to the random initialization instead of the mask; the resulting network remains functionally equivalent. In our work, we chose to permute the mask as we did not want to modify the new random init.
>
>
> > Can the authors clarify what they mean by rewind points? Is it simply the training iteration at which the masks were applied to various dense networks ?
>
> That's correct! Rewind points is the training epoch at which sparse mask is applied to the dense model. We will make it more clear in the paper.
>
> > Is it also the point at which the weights were rewinded during IMP while obtaining the original mask ($\textbf{m}_A$) resulting in different masks for different rewind points?
>
> We used  IMP-FT, which does not use weight rewind to obtain sparse masks. We preferred IMP-FT over IMP with weight rewinding because IMP-FT is computationally less expensive, which allowed us to conduct extensive experiments. In our experiments, we obtain a mask from IMP-FT and apply the mask to the dense model at different rewind points.
>
> > in Figure 4, is the IMP used along with weight rewinding or is IMP-FT used?
>
> Since Paul et al. used IMP (with weight rewinding) for their analysis [1], which suggested only successive IMP iterations are linearly mode-connected, we used IMP (with weight rewinding) in Fig 4. to show that all sparse networks found in each iteration of IMP and dense networks are linearly mode-connected once the variance collapse is taken into account. We decided to use IMP with weight rewinding in Fig 4. for a fair and direct comparison to observations made in [1].
>
> > If IMP-FT was used then does the linear mode connectivity exist between the dense network and the sequential winning tickets of lower density?
>
> We conducted an additional experiment to confirm that linear mode connectivity exists between the dense network and sparse models at each iteration of IMP-FT. We appreciate your attention to detail; this additional analysis provides valuable insights that strengthen our paper's findings. (https://imgur.com/a/2l7khkN)
>
> > The results show a clear trend as the width of the networks increases. The authors show that this is due to better LMC. Can this be simply because the permutation matching is better because each unit has a larger pool of units to match from ?
>
> Yes, that’s correct. We have the same intuition as yours that wider models (with $\ell$ layers, each of width $n$) have more possible permutations (up to $(n!)^{\ell}$), which makes matching two models more accurate. We show this by increasing the model width and comparing the LMC. As observed in Fig.3, on expanding the model width, LMC becomes better, suggesting permutation matching is better for wider models. We will add this explanation/intuition to the final manuscript.
>
> > Additionally, the variance in the unit activations can be studied, larger width may result in lower variance and better match.
>
> This is an interesting insight that could explain why activation (permutation) matching works better for wider models. We analyzed the variance across all layers and observed that the variance for each layer significantly decreases with increasing the width. For example, variance for the *second conv layer in third block* decreases from *0.012* to *0.0001* on increasing width from 1 to 4. Other layers follow the same trend. We will add plots for all the layers in the manuscript.
>
> We hope that we have answered all your questions/concerns; we would really appreciate it if you could update your final score!
>
> [1]  Paul et al., Unmasking the Lottery Ticket Hypothesis: What's Encoded in a Winning Ticket's Mask

---

> > ### Comment · Reviewer_oZBY · 2025-04-03
> >
> > I thanks the authors for the detailed response and for generating additional results. Many of my concerns were addressed.
> >
> > For the mask reuse across the datasets, I request the authors to also add results for naive mask application to the original model at initialization and trained on the new dataset. This may provide an upper bound and allow for clarity regarding the methods use.

---

> > > ### Author Response · Authors · 2025-04-08
> > >
> > > Thank you for this suggestion! We agree that having an upper bound will provide more insight. We have added results on LTH, which will serve as an upper bound (https://imgur.com/a/PUKmoiQ). We will continue running experiments and plan to add more results on mask transfer (masks obtained on ImageNet and tested on the Places365 dataset) to provide more extensive results for the final version of the manuscript.
> > >
> > > We appreciate all of your insights. The additional experiments present more evidence for the effectiveness of our method and further strengthen the contribution of our paper. We thank you for your thorough review process and believe we have now addressed all of your concerns, including the addition of this final experiment. Our work provides novel insights into sparse training from scratch, as you noted, which would be valuable to the efficient ML community, particularly as the field continues to seek methods to reduce computational costs in training large models.
> > >
> > > We would greatly appreciate it if you could update your score if you are satisfied with our rebuttal and the additional experiments. Thank you!

---

### Decision · Program_Chairs · 2025-05-01

**Decision:**

Accept (poster)

**Comment:**

The paper proposes a method to enhance the generalization of Lottery Ticket Hypothesis (LTH) sparse masks by aligning them via permutation to the optimization basin of a new random initialization. While one reviewer raised concerns about clarity, technical novelty, and the strength of claims—particularly regarding initialization versus rewind points—these were addressed effectively in the rebuttal through additional experiments and clarifications. Two other reviewers strongly supported the acceptance of the paper, highlighting its novel insights, empirical robustness across multiple datasets (CIFAR-10/100, ImageNet), and relevance to the sparse training community.

The authors convincingly demonstrated that permuted masks significantly improve performance compared to naïve application. Although computational efficiency and generalization to arbitrary random initializations remain areas for future improvement, the consensus is that the paper makes valuable contributions and addresses important questions in sparse neural network training. Overall, AC recommends acceptance.